# Attribution of N$_2$O sources in a grassland soil with laser spectroscopy based isotopocule analysis

Erkan Ibraim[1,3], Benjamin Wolf[2], Eliza Harris[4], Rainer Gasche[2], Jing Wei[1], Longfei Yu[1], Ralf Kiese[2], Sarah Eggleston[1], Klaus Butterbach-Bahl[2], Matthias Zeeman[2], Béla Tuzson[1], Lukas Emmenegger[1], Johan Six[3], Stephan Henne[1], and Joachim Mohn[1]

[1]Empa, Swiss Federal Laboratories for Materials Science and Technology, Laboratory for Air Pollution & Environmental Technology, CH-8600 Dübendorf, Switzerland

[2] Karlsruhe Institute of Technology, Institute of Meteorology and Climate Research (IMK-IFU), D-82467 Garmisch-Partenkirchen, Germany

[3]ETH-Zürich, Swiss Federal Institute of Technology, Department of Environmental Systems Science, CH-8092 Zürich, Switzerland

[4]University of Innsbruck, Institute of Ecology, Sternwartestrasse 15, A-6020 Innsbruck, Austria

*Correspondence to*: E. Ibraim (erkan.ibraim@empa.ch)

## Abstract

Nitrous oxide (N$_2$O) is the primary atmospheric constituent involved in stratospheric ozone depletion and contributes strongly to changes in the climate system through a positive radiative forcing mechanism. The atmospheric abundance of N$_2$O has increased from 270 ppb during the pre-industrial era to approx. 330 ppb in 2018. Even though it is well known that microbial processes in agricultural and natural soils are the major N$_2$O source, the contribution of specific soil processes is still uncertain. The relative abundance of N$_2$O isotopocules ($^{14}N^{14}N^{16}O$, $^{14}N^{15}N^{16}O$, $^{15}N^{14}N^{16}O$ and $^{14}N^{14}N^{18}O$) carries process-specific

information and thus can be used to trace production and consumption pathways. While isotope ratio mass spectroscopy (IRMS) was traditionally used for high-precision measurement of the isotopic composition of N$_2$O, quantum cascade laser absorption spectroscopy (QCLAS) has been put forward as a complementary technique with the potential for on-site analysis. In recent years, preconcentration combined with QCLAS has been presented as a technique to resolve subtle changes in ambient N$_2$O isotopic composition.

From the end of May until the beginning of August 2016, we investigated N$_2$O emissions from an intensively managed grassland at the study site Fendt in Southern Germany. In total, 612 measurements of ambient N$_2$O were taken by combining preconcentration with QCLAS analyses, yielding $\delta^{15}N^\alpha$, $\delta^{15}N^\beta$, $\delta^{18}O$ and N$_2$O concentration with a temporal resolution of approximately one hour and precisions of 0.46 ‰, 0.36 ‰, 0.59‰ and 1.24 ppb, respectively. Soil $\delta^{15}N$-NO$_3^-$ values and concentrations of NO$_3^-$ and NH$_4^+$ were measured to further constrain possible N$_2$O-emitting source processes. Furthermore,

the concentration footprint area of measured N$_2$O was determined with a Lagrangian particle dispersion model (FLEXPART-COSMO) using local wind and turbulence observations. These simulations indicated that night-time concentration observations were largely sensitive to local fluxes. While bacterial denitrification and nitrifier denitrification were identified as the primary N$_2$O-emitting processes, N$_2$O reduction to N$_2$ largely dictated the isotopic composition of measured N$_2$O. Fungal denitrification and nitrification-derived N$_2$O accounted for 34 - 42 % of total N$_2$O emissions and had a clear effect on the

measured isotopic source signatures. This study presents the suitability of on-site N$_2$O isotopocule analysis for disentangling source and sink processes in-situ and found that at the Fendt site bacterial denitrification/ nitrifier denitrification is the major source for N$_2$O, while N$_2$O reduction acted as a major sink for soil produced N$_2$O.

## 1. Introduction

Nitrous oxide ($N_2O$) is the third most important greenhouse gas (GHG), accounting for 6 % of the total anthropogenic radiative forcing (Myhre et al., 2013), and is thus far the dominant stratospheric ozone depleting substance emitted in the 21st century (Ravishankara et al., 2009). Its globally averaged atmospheric concentration has increased since the preindustrial era from approximately 270 ppb (parts-per-billion, $10^{-9}$ mole mole$^{-1}$) at an average rate of 0.2 – 0.3% yr$^{-1}$ and reached 328.9 $\pm$ 0.1 ppb in 2016 (Prinn, 2016; WMO and GAW, 2016). While it is well known that natural and agricultural soils are the major $N_2O$ sources on a global scale, the relative contributions of individual microbial and abiotic $N_2O$ production and consumption pathways remain largely uncertain because different $N_2O$-producing and -consuming processes are active simultaneously in a soil. Until now, there were no direct methods that allow for the attribution of an emitted amount of $N_2O$ to a given process in the field (Solomon et al., 2007; Billings, 2008; Butterbach-Bahl et al., 2013). However, a detailed understanding of the temporal and spatial variations in $N_2O$ emissions and controlling processes is required to develop mitigation strategies and to better achieve emission reduction targets (Nishina et al., 2012; Cavigelli et al., 2012; Herrero et al., 2016; Decock et al., 2015). Atmospheric $N_2O$ isotopic composition provides important information about $N_2O$ production and consumption processes because distinct microbial and abiotic process pathways exhibit characteristic isotopic signatures (Toyoda et al., 2017; Decock and Six, 2013b; Verhoeven et al., 2018; Denk et al., 2017). Apart from $^{14}N^{14}N^{16}O$, representing 99 % of total atmospheric $N_2O$, the three most abundant $N_2O$ isotopocules are $^{14}N^{15}N^{16}O$ ($^{15}N$ at central $\alpha$ position), $^{15}N^{14}N^{16}O$ ($^{15}N$ at terminal $\beta$ position) and $^{14}N^{14}N^{18}O$ (Toyoda and Yoshida, 1999; Kato et al., 1999). Abundances of isotopocules are usually reported in the $\delta$-notation in per mil (‰) as $\delta^{15}N^{\alpha}$, $\delta^{15}N^{\beta}$, $\delta^{18}O$, calculated according to the equation (1):

$$\delta X = (R_{\text{sample}} - R_{\text{standard}})/\, R_{\text{standard}} \tag{1}$$

where $X$ denotes $^{15}N^{\alpha}$, $^{15}N^{\beta}$ or$^{18}O$ and $R$ refers to $^{14}N^{15}N^{16}O$ / $^{14}N^{14}N^{16}O$, $^{15}N^{14}N^{16}O$ / $^{14}N^{14}N^{16}O$ or $^{14}N^{14}N^{18}O$ / $^{14}N^{14}N^{16}O$, respectively, in a sample or standard (Toyoda and Yoshida, 1999; Brenninkmeijer and Röckmann, 1999; Werner and Brand, 2001). The international isotope reference scale for $^{15}N$ / $^{14}N$ is atmospheric $N_2$ (AIR-$N_2$) and for $^{18}O$ / $^{16}O$ Vienna Standard Mean Ocean Water (VSMOW). Thermal decomposition of isotopically characterized ammonium nitrate ($NH_4NO_3$) has been suggested as an approach to link the position-dependent nitrogen isotopic composition of $N_2O$ to AIR-$N_2$ (Toyoda and Yoshida, 1999; Mohn et al., 2016). The total $^{15}N$ content is usually reported as bulk $^{15}N$ content ($\delta^{15}N^{\text{bulk}}$) according to equation (2):

$$\delta^{15}N^{\text{bulk}} = (\delta^{15}N^{\alpha} + \delta^{15}N^{\beta}) \, / \, 2 \tag{2}$$

while the site preference (SP) is used to denote the intramolecular $^{15}N$ distribution according to the equation (3):

$$SP = \delta^{15}N^{\alpha} - \delta^{15}N^{\beta} \tag{3}$$

The established technique for the analysis of $N_2O$ isotopic composition is isotope-ratio mass-spectrometry (IRMS) (Toyoda and Yoshida, 1999), which is very sensitive and capable of providing highly precise analytical results (Toyoda and Yoshida, 2016). However, IRMS instruments are usually not suitable for field deployment. Recently, quantum cascade laser absorption spectroscopy (QCLAS) (Waechter et al., 2008; McManus et al., 2015), cavity ring-down spectroscopy (CRDS, Erler et al., 2015), and off-axis cavity output spectroscopy (OA-ICOS, Wassenaar et al., 2018) were introduced as alternatives for greenhouse gas (GHG) stable isotope analysis, with the capability for real-time, on-site analysis even at remote locations (Tuzson et al., 2011; Wolf et al., 2015; Eyer et al., 2016; Röckmann et al., 2016). Another advantage of spectroscopic techniques is their ability for direct selective analysis of intra-molecular isotopic isomers (isotopomers) such as $^{14}N^{15}N^{16}O$ and $^{15}N^{14}N^{16}O$, while the determination of the SP using IRMS is only possible via a detour of measuring $\delta^{15}N$-$NO^+$ in combination with $\delta^{15}N^{\text{bulk}}$ and a correcting for scrambling (Toyoda et al., 1999). Several studies have successfully applied QCLAS and CRDS for $N_2O$ isotope analysis in laboratory and field incubation experiments (Koster et al., 2013; Yamamoto et al., 2014;

Erler et al., 2015; Mohn et al., 2013; Winther et al., 2018), and more recently to analyse diurnal and seasonal isotopic variations in ambient $N_2O$ (Mohn et al., 2012; Toyoda et al., 2013; Wolf et al., 2015; Harris et al., 2017). The isotopic composition of $N_2O$ emitted from soils can be extracted from ambient air measurements using traditional two end-member mixing models, i.e. the "Keeling plot" approach (Keeling, 1961) or the Miller–Tans approach. While the Keeling plot approach requires stable background conditions, the Miller-Tans approach is also applicable if the stable background requirement is violated (Miller and Tans, 2003). However, the spatial attribution of the extracted $N_2O$ isotopic composition has to date been neglected because atmospheric transport and turbulence needs to be considered.

The bulk isotopic composition of $N_2O$ produced by biogeochemical source processes, i.e. $\delta^{15}N^{bulk}$ and $\delta^{18}O$, is controlled by fractionation during $N_2O$ production, the isotopic composition of $N_2O$ precursors (i.e., $NH_4^+$, $NO_2^-$, $NO_3^-$ and $H_2O$), and $N_2O$ reduction. In contrast, the difference in $^{15}N$ substitution between the central and terminal position within the $N_2O$ molecule (SP) is independent of the precursor's isotopic composition and characteristic for specific reaction mechanisms or enzymatic pathways (Sutka et al., 2006). Therefore, SP provides distinct process information, which can be determined by pure culture studies and chemical reactions under laboratory conditions (Heil et al., 2014; Wei et al., 2017b; Toyoda et al., 2005). Decock and Six (2013a) and Toyoda et al. (2017) summarized that $N_2O$ from hydroxylamine ($NH_2OH$) oxidation, fungal denitrification and abiotic $N_2O$ production on the one hand and $N_2O$ originating from nitrifier-denitrification and denitrification on the other hand display distinct SP values of $32.8 \pm 4.0$ ‰ and $-1.6 \pm 3.8$ ‰, respectively. Accordingly, SP values of $N_2O$ from mixed microbial communities/ abiotic processes, may display large variations depending on the prominent reaction pathway and the respective study conditions.

With this study, we aim to improve the understanding of the temporal dynamics of $N_2O$ isotopic composition, and to identify the relative contribution of the dominant $N_2O$ producing and consuming microbial processes under field conditions. To achieve this, we i) applied a revised coupled TRace gas EXtractor (TREX) and a QCLAS-based instrumentation (TREX-QCLAS, Ibraim et al., 2018) for the first time during a field campaign for in-situ analysis of $N_2O$ isotopocules from ambient air samples, ii) compared two approaches for the calculation of the isotopic composition of $N_2O$ emitted from soils, namely the Keeling plot versus the Miller-Tans approach, iii) include the isotopic composition of a $N_2O$ precursor, nitrate ($NO_3^-$), to support the identification of dominant processes and iv) use local turbulence and wind profile measurements to outline the spatial extent for which the determined isotopic compositions of soil emitted $N_2O$ are representative.

## 2. Material and Methods

### 2.1 Characterization of the research site Fendt

#### 2.1.1 Study site

The TERENO-preAlpine Observatory (Kiese et al., 2018) research site Fendt (De-Fen), a typical montane grassland south of Munich (Germany) is situated at 595 m a.s.l. and has an annual mean temperature of 8.9 °C with 960 mm mean annual precipitation. The site is intensively managed, which includes up to five times of cutting per year for fodder production followed by manure application as well as occasional cattle grazing (Zeeman et al., 2017). Soil characteristics of the site are given in Table 1. These measurements were carried out between 29 May and 03 August 2016 as part of the ScaleX 2016 campaign (Wolf et al., 2017; https://scalex.imk-ifu.kit.edu/). During the measurement period, management activities included one cut (04 July 2016) and one manure application event (12 July 2016) with a load of 43.7 kg N ha$^{-1}$, of which 20 and 23.7 kg were in the form of organic and ammonium-N, respectively (Raiffeisen Laborservice, Ormont, Germany). The average footprint area for $N_2O$ flux and isotope measurements is given in Figure 7

### 2.1.2 Environmental conditions

Rainfall was determined using four precipitation gauges (Rain collector, Davis instruments, Hayward, CA) as indicated in Figure 1 with triangles. The soil temperature was monitored at three locations across De-Fen (red squares in Figure 1) at three depths (5 cm, 10 cm and 15 cm) using PT100 sensors (IMKO, Ettlingen, Germany). Soil water content was determined within the area (locations are indicated by the dashed square in Figure 1) with five ThetaML2x probes (Delta-T Devices, Cambridge, UK), which integrate soil water content over a soil depth of 0 - 6 cm. Water filled pore space (WFPS) was calculated based on measured volumetric water contents and soil characteristics (Kiese et al., 2018). The atmospheric turbulence statistics were determined using the permanently installed micrometeorological instrumentation (Kiese et al 2018) and additional sonic anemometers installed at 6 m and 9 m above the ground. Vertical wind profiles were determined up to 1000 m above the ground in 20 m intervals using Doppler wind-lidar systems (StreamLine, Halo Photonics, Worcestershire, United Kingdom).

### 2.1.3 Concentrations of soil extracted $NH_4^+$ and $NO_3^-$ and $\delta^{15}N$-$NO_3^-$

Soil samples (approx. 150 g, 2-7 cm depth) were collected twice per week in a sampling grid (mesh size 70 m) spanning the whole measuring area at the De-Fen site (dashed square Fig. 1; Wolf et al., 2017), extracted with 1M potassium chloride (KCl, Merck KGaA, Darmstadt, Germany) and stored at -18 °C. After the manure application, sampling was increased to daily time intervals (12 July 2016 – 15 July 2016), followed by further sampling on 19, 21 and 27 July 2016. The concentrations of $NH_4^+$ and $NO_3^-$ were determined colorimetrically using a spectrophotometer (AGROLAB Agrarzentrum GmbH, Germany).

For 116 out of 298 soil extracts described above, $\delta^{15}N$-$NO_3^-$ was also analysed. This subset of samples was collected at the sampling nodes in the vicinity of the flux chambers and the TREX-QCLAS sample inlet. Soil extracts and 14 KCl blanks were analysed for $\delta^{15}N$-$NO_3^-$ at the Stable Isotope Facility of the University of California Davis, USA using the bacterial denitrification assay (Sigman et al., 2001; Casciotti et al., 2002). The reference materials USGS 32, USGS 34, and USGS 35, as supplied by NIST (National Institute of Standards and Technology, Gaithersburg, MD) were used for data correction and additional laboratory reference materials were included to monitor and correct for instrumental drift and linearity. The standard deviation for repeated measurements of reference material was < 0.2 ‰.

## 2.2 Measurements of soil $N_2O$ fluxes

Soil $N_2O$ flux rates ($f(N_2O)$) were obtained using five replicated opaque static flux chambers coupled with a gas chromatograph with an electron capture detector (GC-ECD) and operated according to a pre-defined schedule. A detailed description of the method can be found for example in Rosenkranz et al. (2006). The chambers were alternately closed and opened for 60 minutes, and each chamber was sampled every 15 minutes, resulting in 4 headspace air measurements per chamber closure time. The chamber dimensions were $50 \times 50$ cm$^2$ and either 15 or 50 cm in height, depending on vegetation height. All flux chambers were deployed south of the mobile laboratory within the dashed square in Figure 1. $N_2O$ fluxes were calculated from the concentration increase over time according to Rosenkranz et al. (2006), taking into account local air pressure and the chamber headspace temperature.

## 2.3 Analysis of $N_2O$ isotopocule by TREX-QCLAS

### 2.3.1 Analytical procedure

The TREX-QCLAS setup used in this study for the $N_2O$ isotope measurements was developed and described in detail by Ibraim et al. (2018), based on a previous system developed for $CH_4$ isotope analysis by Eyer et al. (2014, 2016). In brief, $N_2O$ from 5 L of ambient air is extracted using the TREX device and purged into the multi-pass (76 m) cell of the spectrometer (CW-QC-TILDAS-76-CS; Aerodyne Research Inc., Billerica, USA) by means of a low flow of synthetic air (20.5 % $O_2$, 79.5

% $N_2$, Messer Schweiz AG, Switzerland). This approach is capable of measuring the four most abundant $N_2O$ isotopic species ($^{14}N^{14}N^{16}N$, $^{14}N^{15}N^{16}O$, $^{15}N^{14}N^{16}O$ and $^{14}N^{14}N^{18}O$) at approx. 90 ppm with an Allan deviation of < 0.1 ‰.

The TREX-QCLAS was operated in an air-conditioned mobile laboratory (22 - 30 °C) situated at the north end of De-Fen (Figure 1). Ambient air was continuously sampled with a flow rate of approx. 900 mL min$^{-1}$ from 2 m above the ground at the Eddy Covariance (EC) tower and transported to the mobile laboratory using a SERTOflex tube (~ 20 m length, 6 mm OD, SERTO AG, Switzerland). Then the sample gas was dried using a nafion drier (PermaPure Inc., USA) and subsequently pressurized to 4.5 bars using a membrane pump (PM25032-022, KNF Neuberger, Switzerland). Downstream of the pump the air was passed through a chemical trap for carbon dioxide ($CO_2$) and residual $H_2O$ removal. After this pre-treatment, the air was passed into the TREX device for $N_2O$ pre-concentration following the procedure as described in Ibraim et al. (2018). Maintenance demand during field application was minimized by successively using a multi-position valve (Valco Instruments Inc., Switzerland) to switch between eight chemical traps for $CO_2$ and $H_2O$ removal (Figure 2). Each of the traps consisted of a stainless steel tube (12 mm OD, 350 mm length) filled with 12 g Ascarite (10 - 35 mesh, Fluka, Switzerland), bracketed with magnesium perchlorate ($Mg(ClO_4)_2$, $2 \times 1.5$ g, Fluka, Switzerland) and silane-treated glass wool (Sigma-Aldrich Chemie GmbH, Switzerland). The $CO_2$ extraction capacity of the Ascarite traps was found to be sufficient for > 500 L at ambient $CO_2$ concentrations (unpublished). To avoid $CO_2$ breakthrough and particularly clogging of the trap under varying $CO_2$ and residual $H_2O$ concentrations, the chemical trap was changed every day.

### 2.3.2 Calibration strategy and data processing

The isotopic composition of ambient air was referenced against a set of standard gases (Table 2) that were periodically measured (Figure 2) to ensure long-term repeatability. The measurement routine was implemented using a customized LabVIEW programme. Initially, two standard gases (S1, S2) were analysed for a two-point delta calibration and a target (T) gas was measured to monitor the data quality (Table 2). While S1 and S2 cover the range of $\delta^{15}N^\alpha$ and $\delta^{15}N^\beta$ values of the sample gas, for $\delta^{18}O$ this is currently confined by the non-availability of suitable standard gases. Nonetheless, the implemented calibration procedure presents current best practice in particular as the linearity of the delta scale for QCLAS measurements was demonstrated already in 2008 (Waechter et al., 2008). This phase was followed by a series of four alternating S1 and ambient air sample (S) measurements. A full analytical cycle yielded 13 measurements, including four ambient air analyses, and required approx. four hours, corresponding to a measurement frequency of approx. 1 ambient air sample per hour.

Data processing was conducted as previously described by Harris et al. (2017) using Matlab (MathWorks, Inc., USA). Abundances of the four isotopocules ($^{14}N^{14}N^{16}O$, $^{14}N^{15}N^{16}O$, $^{15}N^{14}N^{16}O$ and $^{14}N^{14}N^{18}O$) were obtained with TDL Wintel (Aerodyne Research Inc., Billerica, USA), and isotope ratios were drift-corrected for changes observed in S1. Specifically, the isotope ratios of S1 were linearly fitted to cell pressure, cell temperature and to the goodness-of-the-TDL-fit. If this linear fit was significant (p-value < 0.05) the correction was applied to all data. These corrections were always relatively small and within the range of 0.05 – 0.2 ‰. In addition, a concentration correction was performed using a linear regression curve determined with S1 diluted in synthetic air. The concentration corrections were -0.20, 0.32 and -0.24 ‰ ppm$^{-1}$ for $\delta^{15}N^\alpha$, $\delta^{15}N^\beta$ and $\delta^{18}O$, respectively. Finally, delta values were calculated from isotope ratios using the two-point delta calibration based on S1 and S2. Since no international standards were available for $N_2O$ isotopes, S1 and S2 were analysed against $N_2O$ standards for which the isotopic composition was assigned at Tokyo Institute of Technology (Tokyo Tech) according to Toyoda and Yoshida (1999). In addition, past and ongoing inter-laboratory comparison measurements on pressurized air indicated a very good agreement with Tokyo Tech results (Mohn et al., 2014; Ostrom et al., 2018).

### 2.4 Source signatures of soil emitted $N_2O$

Source signatures of soil-emitted $N_2O$ were interpreted using the Keeling plot approach (Keeling, 1958). Each analysis started at 7 pm on day *n* and lasted until 6 am on the consecutive day *n+1* local time (UTC +1). This procedure yielded 30 Keeling

plot derived source signatures. The uncertainty of the source signatures was assessed based on the measured isotope delta values and $N_2O$ concentrations using a Monte-Carlo model with 200 iterations. A benchmark value of 10 ‰ for the SP standard deviation was chosen as a criterion to distinguish valid measurements, finally leading to 12 $N_2O$ accumulation events.

For comparison, the source signatures were also calculated with the Miller and Tans (2003) approach. An in-depth description of the implementation of the Miller-Tans method is provided by Harris et al. (2017). In brief, first, a baseline is determined by averaging the data points in the lowest 5 % of the diurnal $N_2O$ concentrations with a 5-day moving window (see SI Figure 3). The same measurement points are also used to find the baseline of the isotope delta values – isotope values are not used to flag the baseline since deviations can be both positive and negative. Subsequently, the Miller-Tans equation (eq. 2 in Harris et al. (2017)) is used to derive the source isotope signatures based on a simple linear regression within a 24-hour moving window. The uncertainty in source isotopic composition is calculated by first propagating measurement errors into all terms used in the Miller-Tans equation and then running 200 iterations assuming a normal distribution of error in all terms.

### 2.5 Footprint analysis with FLEXPART – COSMO simulations

The Lagrangian Particle Dispersion Model FLEXPART (Stohl et al., 2005) was adapted for input from the numerical weather prediction model COSMO (Brunner et al., 2012, Oney et al., 2015 and Henne et al., 2016) and was used on a site-scale to determine the concentration footprint of our observations. For this purpose the model was adapted by locally nudging wind profiles and micro-meteorological observations at De-Fen into the COSMO model output. The latter was taken from the operation analysis and forecast runs by MeteoSwiss with a spatial resolution of approximately 1 km × 1 km. Into these model fields observed profiles of the wind vector (composite of 2.5 m and 9 m sonic anemometer) were locally nudged using a tricubic nudging kernel with a width of 3 km, hence influencing approximately 3 grid cells around the observational site (further related information is provided by Wolf et al. (2017)). Turbulence statistics (friction velocity, Monin-Obukhov length) required by FLEXPART were taken from the observations and locally replaced the COSMO-simulated values. The effect of the nudging procedure was strongest at night and under stable boundary layer conditions, which COSMO often fails to reproduce correctly. FLEXPART was run in backward mode, tracing released model particles 24 h and generating hourly surface source sensitivities ($\tau_{50}$ (s m$^3$ kg$^{-1}$); also called concentration footprint) for the location of the $N_2O$ isotope observations. Source sensitivities were calculated on a regular longitude-latitude grid around the De-Fen site (47.825 – 47.845 °N and 11.50 – 11.51 °E) with a resolution of approximately 50 m × 50 m and for model particles from the surface to 50 m above the ground, the latter of which was also the defined minimum of the model boundary layer height. Multiplication of the source sensitivities with a surface flux and summation over the whole model domain and time of the backward integration yields the concentration increment during the period of simulation. The map of source sensitivities was used as an indicator of the extent of the observed $N_2O$ source. Average source sensitivities were calculated for the 12 accumulation events between 6 pm and 6 am the next day.

## 3. Results

### 3.1 $N_2O$ fluxes and soil parameters

The initial phase of the measurement campaign (10 May 2016 – 21 June 2016) was characterized by low ambient air and soil temperatures (13.5 and 15.6 °C, respectively) along with high precipitation and high WFPS values (> 5 mm d$^{-1}$ and > 95 %, respectively, between 10 – 21 June; Figure 3). Soil extracted $NH_4^+$ and $NO_3^-$ values in this period were 0.27 to 8.32 mg N l$^{-1}$ and 0.12 to 3.15 mg N l$^{-1}$, respectively. This period was also characterized by the lowest $N_2O$ flux rates ($f(N_2O)$), i.e. the mean $f(N_2O)$ of all five chambers was below 70 µg N m$^{-2}$ h$^{-1}$. After 21 June the $N_2O$ fluxes increased, reaching a maximum of approx. 450 µg N m$^{-2}$ h$^{-1}$ on 24 and 25 June. $f(N_2O)$ followed a diurnal pattern with slightly higher emissions during the day but also higher nocturnal $f(N_2O)$ values compared to the initial phase of the campaign. Thereafter $f(N_2O)$ decreased to around 200 µg N m$^{-2}$ h$^{-1}$ on 29 June, before it began to steadily rise from 30 June to 12 July. After the cutting event on 4 July, $NO_3^-$

concentrations increased, while $NH_4^+$ remained unaffected. In contrast, after the manure application on 12 July, the concentration of $NH_4^+$ increased immediately, while $NO_3^-$ only accumulated slowly over the course of the following week. In this period $N_2O$ daytime emissions also peaked at > 900 µg N m$^{-2}$ h$^{-1}$ followed by a period of variable $N_2O$ fluxes with very low but also very high emission rates, for example 17 July and 24 July at 290 and 2400 µg N m$^{-2}$ h$^{-1}$, respectively. Two weeks after the manure application the concentrations of $NH_4^+$ and $NO_3^-$ and $N_2O$ fluxes were comparable to the period prior to manure application and cutting.

## 3.2 Ambient $N_2O$ concentrations and isotopic variations

Figure 4 shows $N_2O$ concentrations and isotopic composition ($\delta^{15}N^\alpha$, $\delta^{15}N^\beta$, $\delta^{18}O$) analysed between 9 June and 23 July in ambient air 2 m above the ground. In total, 612 air sample measurements (S), 150 target gas (T), 1783 anchor gas (S1) and 164 calibration gas (S2) measurements were performed (concentrations and isotopic composition of T, S1 and S2 are given in Table 2). The data gap between 27 June and 8 July was caused by a hard disk failure of the system computer. The standard deviation for repeated in-situ T measurements (undergoing identical treatment compared to S) was 0.46 ‰, 0.36 ‰, 0.59‰ and 1.24 ppb, for $\delta^{15}N^\alpha$, $\delta^{15}N^\beta$, $\delta^{18}O$ and $N_2O$ concentrations, respectively.

Apart from a small nocturnal $N_2O$ concentration increase on 11 June, no clear variations in ambient $N_2O$ were observed in the first three weeks of the campaign, which is in accordance with the lowest soil $N_2O$ fluxes, as described above. On 21 June the onset of a diurnal pattern with nocturnally enhanced $N_2O$ concentrations accompanied by co-varying $\delta^{15}N^\alpha$, $\delta^{15}N^\beta$ and $\delta^{18}O$ values was observed. Mean $N_2O$ concentrations were $331.62 \pm 1.41$ ppb during the day and elevated at night with a maximum of 429 ppb observed on 23 June. During the day, mixed surface-layer isotopic compositions of $N_2O$ were $15.22 \pm 0.42$ ‰, $-2.78 \pm 0.34$ ‰, and $45.88 \pm 0.43$ ‰ for $\delta^{15}N^\alpha$, $\delta^{15}N^\beta$ and $\delta^{18}O$, respectively, thus yielding SP and $\delta^{15}N^{bulk}$ values of $17.95 \pm 0.15$‰ and $6.28 \pm 0.30$ ‰, respectively.

The nocturnal increase of $N_2O$ concentrations was accompanied by a decrease in $\delta^{15}N^\alpha$ and $\delta^{15}N^\beta$, while $\delta^{18}O$ values generally increased at higher $N_2O$ concentrations, but also showed the opposite behaviour for some events. The most extreme $\delta$-values were 8.98 ‰, -9.66 ‰ and 50.61 ‰ for $\delta^{15}N^\alpha$, $\delta^{15}N^\beta$ and $\delta^{18}O$. Compared to the background values, this results in a difference of 6.24 ‰, 6.88 ‰ and 4.73 ‰ for $\delta^{15}N^\alpha$, $\delta^{15}N^\beta$ and $\delta^{18}O$, respectively.

## 3.3 Source signature of soil emitted $N_2O$ and precursors

Source signatures of soil-emitted $N_2O$ at De-Fen were calculated using the Keeling plot method (Keeling 1961, 1958) and the Miller-Tans method (Miller and Tans 2003), as shown in Figure 5. For periods complying with the quality criteria defined for the Keeling plot analysis, results of the two independent techniques agreed reasonably well, as shown in the correlation diagrams in Figure 5. Keeling plot-derived $\delta^{15}N^{bulk}$, $\delta^{18}O$ and SP values varied between -32.5 ‰ and -1.2 ‰, 38.0 ‰ and 65.0 ‰, and 8.4 and 36.8 ‰, respectively; the Miller–Tans analysis resulted in similar source signatures of -29.6 ‰ to 20.3 ‰ ($\delta^{15}N^{bulk}$), 40.7 ‰ to 84.9 ‰ ($\delta^{18}O$) and 5.1 ‰ to 35.0 ‰ (SP) for the same period. The results of the Miller-Tans method were rather scattered for periods when small changes in $N_2O$ concentrations and $N_2O$ isotopic composition precluded Keeling plot analysis (i.e. prior to 22 June). Values of individual Keeling plot derived source signatures can be found in Table 3.

The $\delta^{15}N$-$NO_3^-$ values ranged from 0.13 to 11.42 ‰. Spatial variations of $\delta^{15}N$-$NO_3^-$ across the De-Fen site were relatively large (Figure 5). In the first week of June $\delta^{15}N$-$NO_3^-$ was rather variable with very low values on 9 June but higher $\delta^{15}N$-$NO_3^-$ in the second week. Thereafter it decreased slowly from approx. 10 ‰ to values close to 0 ‰. After the manure application on 12 July a continuous increase of $\delta^{15}N$-$NO_3^-$ was observed, reaching a maximum of approx. 8 ‰ around 24 July.

## 4. Discussion

### 4.1 N$_2$O fluxes and WFPS

Throughout the measurement campaign, the N$_2$O flux rates were between 70 and 2400 µg N m$^{-2}$ h$^{-1}$ at De-Fen, and thus of a similar order of magnitude as reported earlier for other intensively fertilized grasslands (Merbold et al., 2014; Wolf et al., 2015; Schäfer et al., 2012). $f$(N$_2$O) showed a clear dependence on the soil water content, with maximum emissions at 90 % WFPS (Figure 6). While for drier soils (WFPS < 60 %) lower but still substantial N$_2$O fluxes were detected, fluxes declined to their lowest values near water saturation, i.e. when WFPS was close to 100%. The observed relationship between $f$(N$_2$O) and WFPS (R$^2$ = 0.92) can be best described with an exponential function with two terms as given by equation 4:

$$(N_2O)_{Fitted} = a \cdot \exp(b \cdot WFPS) + c \cdot \exp(d \cdot WFPS) \qquad (4)$$

where the coefficients are best approximated by $a$ = -5.09e-06, $b$ =0.19, $c$ =15.86 and $d$ = 0.04. This relationship is a strong indicator that the activity of the main source process increases with the soil water content, which is characteristic for denitrification and nitrifier-denitrification (Wrage et al., 2001; Decock and Six, 2013a). Furthermore, the decline of N$_2$O fluxes at very high WFPS values is in line with this interpretation, because the last step of the denitrification pathway, N$_2$O reduction to N$_2$, is only active under anoxic conditions. This shift from nitrification-dominated to denitrification-dominated N$_2$O production with increasing WFPS should be reflected in the isotopic signature of the residual N$_2$O. Indeed, there is a tendency towards high SP values under low (indicating higher nitrification contribution) and high WFPS values (indicating higher N$_2$O reduction to N$_2$ rates) (Figure 6). The peak $f$(N$_2$O) was observed on 23 July, a day after a severe precipitation event. The N$_2$O emission rate of this peak event was 2415 µg N m$^{-2}$ h$^{-1}$ (average of five replicate flux chambers). Unfortunately, this event cannot be discussed in terms of N$_2$O isotopocules due to termination of TREX-QCLAS measurements after 22 July 2016.

### 4.2 On-site performance of TREX-QCLAS

The short term repeatability over 10 target gas (T) measurements was 0.25 ‰, 0.31 ‰, 0.30 ‰ and 0.25 ppb for $\delta^{15}N^{\alpha}$, $\delta^{15}N^{\beta}$, $\delta^{18}O$ and N$_2$O concentration, respectively. This is sufficient to track changes in ambient N$_2$O close to emission sources as described in this study and superior to most IRMS and laser spectrometer systems (Mohn et al., 2014), but slightly inferior to laboratory experiments using the same system (Ibraim et al., 2018) or earlier versions of preconcentration – QCLAS based approaches (Mohn et al., 2012; Harris et al., 2014; Wolf et al., 2015). The slightly lower repeatability was due to a more compact spectrometer design, which allowed for the integration of the system in a 19-inch rack at the cost of a higher optical noise level and larger drifts due to the harsher conditions in the mobile lab, i.e. higher temperature variations and vibrations.

### 4.3 Variability of N$_2$O concentrations and isotopic composition above De-Fen

During the day, the atmospheric boundary layer (ABL) and the lowest part of the ABL (surface layer) are well mixed due to turbulence arising from buoyancy and wind shear (Ibbetson, 1994). At night, stable stratification attenuates vertical mixing processes, also leading to generally lower horizontal wind speeds. Both entail accumulation of local soil-emitted N$_2$O in the surface layer. For this reason, daytime N$_2$O concentrations and isotopic composition mostly reflect the atmospheric background, while the nighttime accumulation reflects the influence of soil-emitted N$_2$O.

Variations in N$_2$O, SP, $\delta^{15}N^{bulk}$ and $\delta^{18}O$ follow a diurnal pattern that is in agreement with the variations of N$_2$O concentrations depicted in Figure 4. Accordingly, average daytime N$_2$O concentrations, $\delta^{15}N^{bulk}$, SP and $\delta^{18}O$ of 331.6 ± 1.41 ppb, 6.28 ± 0.30 ‰, 17.95 ± 0.15 ‰ and 45.54 ± 0.27 ‰, respectively, are in agreement with background measurements at other sites, such as Dübendorf, Switzerland (N$_2$O: 325.8 ± 3.3 ppb, $\delta^{15}N^{bulk}$: 6.53 ± 0.14 ‰, SP: 17.95 ± 0.40 ‰, $\delta^{18}O$: 44.41 ± 0.13 ‰ (Harris et al., 2017)), or Hateruma Island, Japan (decadal mean values for the northern hemisphere of $\delta^{15}N^{bulk}$: 6.65 ‰, SP: 18.44 ‰, $\delta^{18}O$: 44.21 ‰ (Toyoda et al., 2013)). Observed changes in N$_2$O concentrations and isotopic composition at night are within

the range of previous studies from agricultural sites (Wolf et al., 2015; Toyoda et al., 2011), but clearly higher than variations measured at 13 m or 95 m above ground in an urban or suburban environment (Harris et al., 2014; Harris et al., 2017).

### 4.3.1    $N_2O$ footprints

At night, within a stable nocturnal boundary layer, vertical wind speeds and hence tracer transport are low, while lateral wind speeds can be high and constituents like $N_2O$ can be transported over larger distances. As a result, $N_2O$ emissions from other land uses or land cover have contributed to the observed $N_2O$ isotopic composition. To assess the influence of other land use / land cover, the concentration footprint calculated with FLEXPART-COSMO was assessed for periods where the Keeling plot and Miller-Tans approaches were applied. The FLEXPART-COSMO simulations indicate that between 15 % and 45 % of the source sensitivity originates from areas within approximately 300 m to 700 m distance to the sample inlet, respectively (isolines in Figure 7). Highest source sensitivities which amounted to 30% of the total sensitivity were calculated for areas predominately covered by grassland or pasture. Although sources outside this local area contributed more than half of the total emissions and included other land cover such as arable land and forest, the impact of individual source areas was smaller by several orders of magnitude, hence having much less impact on the isotopic source signature. While more than 95 % of the area covered by the 15 % isopleth (bold isolines in Figure 7) corresponds to grasslands, the residual 5 % belongs to a wetland to the northeast of the De-Fen (Figure 7). Furthermore, the 30 and 45 % isopleth's surfaces include approximately 20 % of mixed forest and 5 % wetland along with around 75 % under grassland, underlining further that sensitivities were highest for grassland emitted $N_2O$.

In addition to the $N_2O$ footprint, the temporal trend of the $N_2O$ concentration at the sampling point was simulated using individual source sensitivities and assuming a homogeneous $N_2O$ flux identical to measured local $N_2O$ fluxes (see section 2.2). Simulated $N_2O$ concentrations were in very good agreement with $N_2O$ concentrations measured by the TREX-QCLAS (SI Figure 10), indicating that the simulated footprint, attributing a substantial part of the emissions to the De-Fen grassland, is representative of the measurement site. Furthermore, $N_2O$ concentration measurements obtained with TREX-QCLAS were in a good agreement with the local $N_2O$ flux measurements (SI Figure 2).

### 4.4  $N_2O$ source signatures and implicated processes

### 4.4.1    Comparison Miller–Tans and Keeling plot techniques

Figure 5 shows the temporal trends of the $N_2O$ source signatures, illustrating the potential of this quasi-continuous dataset to identify process changes induced by management events or changing environmental parameters. The dataset also enables a direct comparison of two approaches for extracting the isotopic composition of $N_2O$ emitted from soils based on surface layer measurements, namely the Keeling plot and the Miller-Tans approach. In the first three weeks of the campaign, i.e. under conditions of low $N_2O$ fluxes, the Keeling plot results did not pass the quality criterion, and the source signatures, i.e., the calculated isotopic composition of $N_2O$ emitted from soil ($\delta^{15}N^{bulk}$, SP, $\delta^{18}O$) derived from the Miller-Tans method showed relatively large uncertainties and amounted to 2.8 – 9.8 ‰, 2.3 – 10.6 ‰ and 4.6 – 12.9 ‰, respectively (shaded areas in Figure 5). Thereafter, $N_2O$ source signatures as estimated with the Keeling plot and Miller–Tans approaches show a comparable trend and mostly agree within the indicated uncertainties without systematic deviations. Overall, the agreement ($R^2$ value) between the Miller–Tans and Keeling plot results is best for $\delta^{15}N^{bulk}$ (0.84), intermediate for SP (0.57) and weakest for $\delta^{18}O$ (0.39) (Figure 5). The weaker correlation for $\delta^{18}O$-$N_2O$ can be explained by a lower analytical data quality as compared to $\delta^{15}N^{bulk}$ and SP, exemplified by a higher standard deviation for repeated measurements of the target gas (0.59 ‰ for $\delta^{18}O$ and 0.41 ‰ for $\delta^{15}N^{bulk}$ and SP). The reasoning behind this effect might be that the calibrated range of $\delta^{18}O$ values (S1, S2) does not cover the isotopic composition of the target and sample gases, because no suitable calibration gas was available. A difference of 7

‰ in $\delta^{18}O$ between the two calibration gases is rather small, leading to a relatively high uncertainty in the respective calibration factors.

The base calculation for both the Keeling plot and Miller-Tans is identical and the two methods would yield identical results if every term was known perfectly. However, the uncertainty term is treated differently in the two approaches. The Miller-Tans approach calculates source signatures for individual sample gas measurements (SI Figure 3) and, thus, may be the better choice when the source process or the background $N_2O$ isotopic composition changes rapidly, i.e. during a 24 hour period. However, the large fluctuations of the source signatures (up to 100 ‰, Figure 5) extracted by the Miller-Tans approach prior to 22 June indicate that the uncertainty estimated for the Miller-Tans approach is too optimistic and needs to be reassessed. In addition, it is noteworthy that the Keeling plot approach as presented here, implicitly considers changes in background $N_2O$ concentration from day to day, since one Keeling plot (comprising both $N_2O$ background and $N_2O$ variations) was carried out per day. Therefore, we conclude that the Keeling plot method remains a robust way of estimating source signatures of $N_2O$ emitted from a predominantly agricultural landscape as the one presented here, where variations in background $N_2O$ compared to source contributions can be neglected and changes in source processes generally occur only on long timescales as a response to changes in environmental conditions (e.g. WFPS).

### 4.4.2    Range of N₂O source signatures

Typical source signatures of biologically produced $N_2O$ are approx. -40 – 0 ‰ and 0 – 40 ‰ for $\delta^{15}N^{bulk}$ and SP, respectively, while $\delta^{18}O$-$N_2O$ are around 40 ‰ and 70 ‰ for $N_2O$ emitted through grasslands or wetlands, respectively (Toyoda et al., 2017). Accordingly, the $\delta^{15}N^{bulk}$ values found in our study are well within literature values of grassland emitted $N_2O$, while the SP values are rather high. Interestingly, the obtained $\delta^{18}O$ values were strongly elevated on some occasions and close to those found by Ostrom et al. (2007) in a pure culture experiment in which approx. 80 % of produced $N_2O$ was reduced to $N_2$. A correlated increase of the SP, $\delta^{15}N^{bulk}$ and $\delta^{18}O$ values, with SP values potentially larger than the endmember value of 32.8 ± 4‰, can be explained by $N_2O$ reduction to $N_2$, which is particularly active under wet and anaerobic soil conditions (Wrage et al., 2004; Lewicka-Szczebak et al., 2017). Thus, isotopic fractionation during partial $N_2O$ reduction must be taken into account in order to apportion isotopic source signatures of soil-emitted $N_2O$ (Lewicka-Szczebak et al., 2017; Verhoeven et al., 2018). The fractionation factors $\varepsilon^{18}O/\varepsilon^{15}N^{bulk}$, $\varepsilon^{18}O/\varepsilon SP$ and $\varepsilon^{15}N^{bulk}/\varepsilon SP$ have been determined in a number of incubation experiments and it has been suggested that their ratios (2.4, 2.8 and 1.2, respectively) may be indicators for $N_2O$ reduction (Koba et al., 2009). It has to be mentioned, however, that fractionation factors may deviate depending on environmental conditions (Koster et al., 2013) or even over the course of a single experiment due to multiple reaction steps involved (Haslun et al., 2018). Furthermore, $\delta^{18}O$-$N_2O$ of denitrification is affected by oxygen exchange between reaction intermediates ($NO_3^-$, $NO_2^-$) and soil water as a function of WFPS (Well et al., 2008; Kool et al., 2011).

### 4.4.3    N₂O source partitioning using SP and Δδ¹⁵Nᵇᵘˡᵏ

An SP-versus-$\Delta\delta^{15}N^{bulk}$ (Figure 8(a)) mapping approach as originally presented by Koba et al. (2009) was used to interpret the Keeling plot-derived source signatures with respect to the possible underlying $N_2O$ producing and consuming processes. Here, $\Delta\delta^{15}N^{bulk}$ denotes the $\delta^{15}N$ difference between the product $N_2O$ and its substrate ($NO_3^-$). While Koba et al. (2009) applied this approach in the framework of a groundwater study where $NO_3^-$ was the only available $N_2O$ substrate, the grassland research site De-Fen showed rather high $NH_4^+$ concentrations (Figure 3(b)). Therefore, the $N_2O$ substrate at De-Fen might be either $NH_4^+$ for $N_2O$ emitted by nitrification (N) and nitrifier-denitrification (ND) or $NO_3^-$ from fungal denitrification (FD) and bacterial denitrification (BD). Within the framework of this study, it was assumed that $\delta^{15}N$–$NH_4^+$ and $\delta^{15}N$–$NO_3^-$ values were in a similar range, i.e. approx. 0 – 15 ‰, in agreement with the literature (Mook, 2002; Holland, 2011). We thus used only the $\delta^{15}N$–$NO_3^-$ values for the substrate isotopic composition. For periods where $N_2O$ emissions were present but no $\delta^{15}N$–$NO_3^-$

values were obtained, the $\delta^{15}N$–$NO_3^-$ values were approximated by linear interpolation. In addition, the concept of Koba et al. (2009) was modified for the two $N_2O$-emitting domains FD/N and BD/ND using literature values as provided in Table 4. For simplicity, in the remaining part of this section the flux-weighted average values of SP and $\Delta\delta^{15}N^{bulk}$ are discussed, while values of individual events can be found in Table 4. Plotting SP vs. $\Delta\delta^{15}N^{bulk}$ revealed that there was a trend of increasing SP

with decreasing $\Delta\delta^{15}N^{bulk}$ values. As indicated in Figure 8 with orange crosses, the flux-averaged SP, $\Delta\delta^{15}N^{bulk}$ and $\delta^{18}O$ values were 23.4, 19.0 and 62.3 ‰, respectively. The slope of the SP-versus-$\Delta\delta^{15}N^{bulk}$ linear regression line of -0.85 (solid red arrow in Figure 8(a)) is in agreement with literature values (-0.83 and -1.1) given by Koba et al. (2009) and Toyoda et al. (2017) for partial $N_2O$-to-$N_2$ reduction. This observed negative slope, which is in contrast to the grey shaded area anticipated for mixing of $N_2O$ produced by BD/ND and FD/N indicates a major contribution of BD/ND and $N_2O$ reduction to $N_2$, the final reaction

step in the anoxic reduction of $NO_3^-$ to $N_2$. The suspected predominance of denitrification agrees with previous field studies presented by Opdyke et al. (2009), Wolf et al. (2015) and Mohn et al. (2012). SI Figure 9 illustrates contributions of FD/N on the total $N_2O$ emissions for individual accumulation events.

A semi-quantitative source partitioning can be calculated assuming average SP (-0.9 ‰) and $\Delta\delta^{15}N^{bulk}$ (18.5 ‰) values for $N_2O$ production by BD/ND and a fixed SP/$\Delta\delta^{15}N^{bulk}$ ratio of -0.83 for $N_2O$ reduction to $N_2$ (Figure 8(a)). Correspondingly,

the simultaneous SP increase and $\Delta\delta^{15}N^{bulk}$ decrease during $N_2O$ reduction to $N_2$ can be interpreted in terms of the $N_2O$/ ($N_2O$+$N_2$) product ratio using the Rayleigh fractionation approach of Mariotti et al. (1981). Accordingly, a 90 % reduction of $N_2O$ translates into an increase in SP by 13.6 ‰ assuming an SP fractionation factor ($\varepsilon$SP) of -5.9 ‰ in accordance with Ostrom et al. (2007). Using a single $\varepsilon$(SP) value is a simplification, however, as fractionation factors might vary e.g. depending on WFPS (Jinuntuya-Nortman et al, 2010) and $N_2O$/($N_2$+$N_2O$) product ratio (Lewicka-Szczebak et al., 2015). A deviation of

source signatures from the SP/$\Delta\delta^{15}N^{bulk}$ line can then be interpreted in terms of addition of $N_2O$ produced by additional processes, e.g. FD/N. This interpretation is supported by the relationship between SP and WFPS (Figure 6). Accordingly, the lowest SP values were found at intermediate to high soil water contents (80 – 90 % WFPS) along with maximum $N_2O$ fluxes, while SP values increased towards lower WFPS values, due to the increasing contribution of nitrification, and towards higher WFPS values, due to increasing $N_2O$ reduction to $N_2$. Furthermore, the fraction of FD/N-derived $N_2O$ increased with $NH_4^+$

fertilization, also in agreement with the literature (Toyoda et al., 2011; Koster et al., 2011).

A semi-quantitative interpretation of isotope signatures can be done assuming average source signature values (SP and $\Delta\delta^{15}N^{bulk}$) and considering two scenarios (see also SI Figure 4): in scenario 1, BD/ND-produced $N_2O$ is partially reduced to $N_2$ and the residual $N_2O$ ($rN_2O$; remaining $N_2O$ after $N_2O$ reduction to $N_2$) is then mixed with $N_2O$ derived from FD/N (path of solid arrows in Figure 8(a)). In scenario 2, $N_2O$ from BD/ND is mixed with FD/N-derived $N_2O$ before a part of

the mixed $N_2O$ is reduced to $N_2$ (path of dashed arrows in Figure 8(a)). While these scenarios result in equal source signatures, they assign a different relative contributions of the processes involved. The respective $N_2O$ to $N_2$ reduction rates can be calculated based on the associated shift in SP, which corresponds to the y component of each of the red arrows in Figure 8(a). For convenience, here we only discuss the reduction rates and source partitioning of the two scenarios for flux-averaged SP and $\Delta\delta^{15}N^{bulk}$ values (23.4 and 19.0, respectively), while those of individual events could be estimated analogously (related

results given in Table 3). Assuming scenario 1, the SP shift caused by $N_2O$ reduction is equal to 18.0 ‰; resulting in a reduction rate of approx. 95 % assuming $\varepsilon$SP = -5.9 ‰. The remaining 5.4 ‰ SP shift can be explained as the result of mixing the $rN_2O$ with FD/N-derived $N_2O$. A 5.4 ‰ SP shift corresponds to approx. 38 % contribution of FD/N-derived $N_2O$ with the residual $N_2O$ emitted by BD/ND. Note that the FD/N contribution is less than 1 % when accounting for the total $N_2O$ production, i.e. the $N_2O$ before partial reduction to $N_2$. In contrast, in scenario 2, the FD/N-derived $N_2O$ is mixed with BD/ND-derived $N_2O$

first. This mixing induces a SP shift of approx. 13.0 ‰, which is given by the y-coordinate of the intersection of the mixing-line and the reduction line of the mean source signatures. However, since no $N_2O$ reduction to $N_2$ has occurred yet at this point,

this shift corresponds to 39 % contribution of FD/N to total $N_2O$ production. The remaining 10.4 ‰ SP shift is then subject to reduction of $N_2O$ to $N_2$, corresponding to approx. 83 % reduction of $N_2O$ to $N_2$.

### 4.4.4     $N_2O$ source partitioning using SP and $\Delta\delta^{18}O(N_2O/H_2O)$

Identification of the processes producing and consuming $N_2O$ was also done using an adapted SP-versus-$\Delta\delta^{18}O(N_2O/H_2O)$ mapping approach (Figure 8(b)) as previously presented by Lewicka-Szczebak et al. (2017). This approach was suggested because the values of $\delta^{18}O$-$N_2O$ from BD/ND and FD/N are less variable than those of $\delta^{15}N$-$N_2O$. The lower variability is indicated by the smaller BD/ND and FD/N boxes in Figure 8(b) compared to Figure 8(a); thus, using this approach reduces the uncertainty of the calculated relative contributions of the different processes as the boxes are used to span the mixing line. $\Delta\delta^{18}O(N_2O/H_2O)$ for denitrification is considered to be constant (Lewicka-Szczebak et al., 2016), in particular under high WFPS associated with close to 100 % oxygen exchange between soil water and reaction intermediates (Kool et al., 2011). The approach was slightly modified using the values presented in Table 4 to match the FD/N and the BD/ND domains according to Figure 8(a) with regard to SP values. In this approach, $\Delta\delta^{18}O(N_2O/H_2O)$ represents the difference between the $\delta^{18}O$ values of the product ($N_2O$) and the substrate ($H_2O$). Since no measurements for $\delta^{18}O$-$H_2O$ were available, we used a value of -8 ‰ in accordance with Xiahong et al. (2009). Values obtained for $\Delta\delta^{18}O(N_2O/H_2O)$ were clearly higher than previously observed in grassland soils (Wrage et al., 2004; Wolf et al., 2015; Snider et al., 2012) but particularly close to $\Delta\delta^{18}O(N_2O/H_2O)$ values from studies related to wetland ecosystems (Toyoda et al., 2017; Snider et al., 2009), likely reflecting the fact that the study site was in the vicinity of a wetland (see section 4.3.1 and Wolf et al., 2017) and often flooded due to extraordinary precipitation events throughout the measurement period.

In the mapping approach suggested by Lewicka-Szczebak et al. (2017), two scenarios are considered to estimate the shift in $N_2O$ isotopic composition due to $N_2O$ reduction to $N_2$. In Figure 8(b), the y-component of the red arrows represents the SP shift that was caused by $N_2O$ reduction to $N_2$. Knowledge of the degree to which SP has been changed due to fractionation during $N_2O$ reduction is a prerequisite for determining the relative contributions of the process groups BD/ND and FD/N using a simple mixing model and the SP values given in Table 4. Scenario 1 assumes that BD/ND-derived $N_2O$ is partly reduced to $N_2$ before mixing with $N_2O$ originating from FD/N, while scenario 2 assumes the reverse order (i.e. first mixing, then $N_2O$ reduction). The two scenarios yield different reduction rates and proportions of BD/ND- versus FD/N-derived $N_2O$, although final $N_2O$ source signatures are identical. A quantitative estimate of source contributions was conducted for the flux averaged mean values of 23.4 and 62.3 ‰ for SP and $\Delta\delta^{18}O(N_2O/H_2O)$ as follows: using scenario 1 (depicted with solid arrows in Figure 8(b)), $N_2O$ reduction to $N_2$ has led to an SP shift of approx. 17.3 ‰, which corresponds to approx. 95 % $N_2O$ reduction. The residual SP shift of 6.1 ‰ would be caused by the mixing of FD/N-derived $N_2O$ with the r$N_2O$, corresponding to approx. 19 % FD/N-derived $N_2O$ compared to BD/ND. The 19 % mentioned here only accounts for the mixing with the r$N_2O$ but not for the initially produced $N_2O$. Taking into account that 95 % of the $N_2O$ initially produced was reduced to $N_2$ reveals that the FD/N contribution to total $N_2O$ production was below 1 %. In contrast, in scenario 2, the FD/N-derived $N_2O$ is mixed into the $N_2O$ pool before $N_2O$ reduction to $N_2$ has occurred. Therefore, approx. 29 % FD/N-derived $N_2O$ is needed to account for a 16 ‰ SP shift in the produced $N_2O$. In this case, the residual SP shift of 9 ‰ is due to $N_2O$ reduction, corresponding to a 79 % reduction rate with $\varepsilon$SP = -5.9 ‰.

### 4.4.5     Comparison of the results obtained with the SP-vs.-$\Delta\delta^{15}N^{bulk}$ and SP-vs.-$\Delta\delta^{18}O(N_2O/H_2O)$ approaches

In summary, the two scenarios lead to different calculated relative amounts of $N_2O$ produced by BD/ND and FD/N as well as the emissions ratio of $N_2O$ to $N_2$. The average contribution of FD/N to the $N_2O$ emissions was 42 and 34 % according to the SP-vs.-$\Delta\delta^{15}N^{bulk}$ and SP-vs.-$\Delta\delta^{18}O(N_2O/H_2O)$ approaches, respectively (distributions given in SI Figure 8, temporal trend given in SI Figure 9). However, regardless of the approach and scenario, the obtained r$N_2O$ values were very low, indicating

that $N_2O$ reduction played a major role. The median of the $rN_2O$ values obtained with the SP-vs.-$\Delta\delta^{15}N(NO_3^-/N_2O)$ approach was 0.02 for scenario 1 and 0.10 for scenario 2. Utilizing the SP-vs.-$\Delta\delta^{18}O(N_2O/H_2O)$ approach, those values were even slightly lower and corresponded to 0.01 in scenario 1 and 0.02 in scenario 2 (SI Figure 5). Interestingly, the two rN2O values calculated for scenario 1 with the two approaches were highly correlated, while those for scenario 2 were not correlated (SI

Figure 5). This indicates that scenario 1 more likely occurred at our site.

The $rN_2O$ values were also compared to the WFPS (SI Figure 6) and to the ambient temperature (SI Figure 7). A positive correlation should be expected between WFPS and the $N_2O$ reduction rates, resulting in a negative correlation between WFPS and $rN_2O$ values. However, observed $rN_2O$ values did not reflect this hypothesis. Similarly, one could expect a positive correlation between rNit (the fraction of measured $N_2O$ originating from fungal denitrification or nitrification, therefore with

high SP values) and $rN_2O$, since the contributions of fungal denitrification and nitrification should be higher under conditions that are disadvantageous for $N_2O$ reduction. However, also this hypothesis was refuted by these results.

Our findings confirm that natural abundance isotope studies of $N_2O$ provide a way to trace $N_2O$ production / destruction pathways, in particular when combined with supportive parameters or isotope modelling approaches (Denk et al., 2017). However, the complexity of $N_2O$ production pathways could not be fully accounted for, in particular abiotic processes; for

example, $N_2O$ production by $NH_2OH$ oxidation (Heil et al., 2014) or $NO_2^-$ reduction (Wei et al., 2017a) were not considered. These reactions yield $N_2O$ with high (34 – 35 ‰) or variable (8 – 12 ‰) SP and might therefore be falsely interpreted as nitrification-derived $N_2O$. In addition, the approach cannot resolve individual processes with high SP, i.e. fungal denitrification versus nitrification, or low SP, i.e. heterotrophic versus nitrifier denitrification, due to overlapping source signature regions. Furthermore, nitrite ($NO_2^-$) and nitric oxide (NO) could have acted as the substrate instead of $NO_3^-$, leading to different

fractionation factors from those incorporated for $NO_3^-$.

### 4.4.6    Effect of manure application on the source signatures

In addition to the mapping approaches discussed above, $N_2O$ source signatures can be interpreted with respect to management events. After the manure application on 12 July and rainfall events in the days thereafter a strong shift to lower SP and $\delta^{15}N^{bulk}$ values was observed (Figure 5). The negative shift in $\delta^{15}N^{bulk}$ might be explained by changes in the isotopic composition of

the applied precursors, by an enhanced fractionation due to higher substrate availability or changes in process conditions (e.g. WFPS, see sections above). However, since SP is considered to be process-specific and substrate-independent (Yoshida and Toyoda, 2000), it should not change as a response to a change in the substrate isotopic composition or by enhanced fractionation. There are two alternative explanations for the lower SP and $\delta^{15}N^{bulk}$ values. The increase in $NH_4^+$ concentration after manure application was followed by an increase of $NO_3^-$ concentration. This indicates a stimulation of nitrification. An

increase of $N_2O$ production due to nitrification would be associated with higher SP values. However, the nitrate produced during nitrification may have been used as substrate for denitrification, given the increase in WFPS due to intensive rainfall events. While $N_2O$ is an obligatory product of denitrification, and only a by-product of nitrification, the $N_2O$ yield of denitrification may have been higher, and the increase of SP due to nitrification may have been outweighed by the decrease of SP due to denitrification. Secondly, $N_2O$ reduction to $N_2$ could be slightly reduced due to an elevated $NO_3^-$ availability (Wang

et al., 2013). A parallel increase in WFPS and $N_2O$ flux rates after the manure application combined with low FD/N fraction in the period 17 July to 22 July supports the hypothesis that both effects might have contributed to a decrease in SP values.

### 5.    Conclusions

Real-time and in-situ $N_2O$ concentration and isotope measurements were successfully performed at a temperate humid grassland site in Southern Germany with a coupled preconcentration technique and quantum cascade laser absorption

spectroscopy (TREX-QCLAS) based method in a two-month period between June and July 2016. Replicate in-situ

measurements of a pressurized air tank demonstrated a short term repeatability of the TREX-QCLAS system of 0.25 ‰, 0.31 ‰, 0.30 ‰ and 0.25 ppb for $\delta^{15}N^\alpha$, $\delta^{15}N^\beta$, $\delta^{18}O$ and $N_2O$ concentration, respectively. Accuracy of results was assured using a two point calibration that entirely spanned the range of obtained $\delta^{15}N^\alpha$ and $\delta^{15}N^\beta$ values, however, did not fully cover the range of obtained $\delta^{18}O$ values. This is current best practice, as no suitable reference gases are available, but might lead to a somewhat

larger uncertainty for $\delta^{18}O$-$N_2O$. Concentrations of soil-extracted $NH_4^+$, $NO_3^-$ and $\delta^{15}N$-$NO_3^-$ values were taken into account to interpret the $N_2O$ measurements. This study provides new insights into the isotopic composition of grassland-emitted $N_2O$ under changing soil environmental and management conditions. Our results support previous observations that bacterial denitrification/ nitrifier denitrification (BD/ND) is the dominant $N_2O$-emitting source in permanent grassland soils. The measured $N_2O$ isotopic composition, in particular the intramolecular isotopic composition, or site preference (SP), can be

explained by taking into account partial $N_2O$ reduction to $N_2$. Two distinct approaches were used to estimate the relative contributions of BD/ND and FD/N as well as the $N_2O$ reduction rates. The average FD/N contribution to the total $N_2O$ emissions was 42 and 34 % with the SP-vs.-$\Delta\delta^{15}N^{bulk}$ and SP-vs.-$\Delta\delta^{18}O$ approaches, respectively, indicating that denitrification dominated the $N_2O$ emissions. $N_2O$ reduction rates were estimated by calculating the residual $N_2O$ fractions ($rN_2O$), i.e. the fraction of remaining $N_2O$ after $N_2O$ reduction to $N_2$ has occurred. Two distinct scenarios were considered for each of the two

approaches, resulting in the four $rN_2O$ values of 0.02, 0.10, 0.02 and 0.01. The low values underline both the dominant role of denitrification in $N_2O$ production at the grassland site and the large extent to which $N_2O$ reduction occurred during the measurement period.

This study demonstrates the suitability of the TREX-QCLAS for in-situ analysis of the isotopic composition of soil-emitted $N_2O$ in terrestrial ecosystems. While the observations presented here integrate $N_2O$ fluxes and thus source processes at the plot

scale, the interpretation of source processes in future studies will be resolved at smaller spatial scales, for example by a combination of TREX-QCLAS with static flux chambers and the implementation of an isotopic biogeochemical soil model. In particular, an approach based on the combination of the TREX-QCLAS method with static flux chambers would allow us to distinguish between the two scenarios (reduction then mixing vs. mixing then reduction) discussed in this study.

### 6.   Acknowledgements

This project was financially supported by the Swiss National Science Foundation within the grant number 200021L_150237 and 200020L_172585 / 1 and the German Research Foundation within grant number BU 1173/15-1 and ZE 1006/2-1. The TERENO-preAlpine Observatory infrastructure is funded by the Helmholtz Association and the Federal Ministry of Education and Research. Pietro Ferretti is acknowledged for his technical support in constructing the chemical trap module. We would like to thank the anonymous reviewers of this article.


## 7. Tables and Figures

### 7.1 List of table captions

### 7.2 Tables

**Table 1 Soil characterization of the research site Fendt. Values are given for the topsoil (0-10 cm) according to Kiese et al. (2018).**

| Soil type | Texture sand/ silt/ clay (%) | Bulk density (g*cm$^{-3}$) | pH (a.u.) | Total nitrogen (%) | Soil organic carbon (%) |
|---|---|---|---|---|---|
| Cambic stagnosol | $27 \pm 2 / 43 \pm 1/ 30 \pm 1.3$ | $1.1 \pm 0.1$ | $5.7 \pm 0.3$ | $0.43 \pm 0.01$ | $3.9 \pm 0.4$ |

**Table 2 Mole fractions and isotopic compositions of standard 1 (S1), standard 2 (S2) and target (T) gas cylinders that were used in this study. $N_2O$ mole fractions were analysed at Empa against standards from commercial suppliers (S1, S2) or from the National Oceanic and Atmospheric Administration/ Earth System Research Laboratory/ Global Monitoring Division (NOAA/ ESRL/ GMD) (T). $N_2O$ isotopic composition was also analysed at Empa against standards previously analysed by Sakae Toyoda/ Tokyo Institute of Technology. The standard gas S1 is used for drift correction, and standard gas S2 for a span-correction of measured δ values. The indicated error is one standard deviation for replicate sample measurements and does not include the uncertainties of the calibration chain.**

| Gas Type | $\delta^{15}N^{\alpha}$ (‰) | $\delta^{15}N^{\beta}$ (‰) | $\delta^{18}O$ (‰) | $N_2O$ mole fraction (ppm) |
|---|---|---|---|---|
| S1 | $15.51 \pm 0.30$ | $-3.25 \pm 0.20$ | $34.97 \pm 0.16$ | $90.15 \pm 0.005$ |
| S2 | $-63.08 \pm 0.78$ | $-59.81 \pm 0.48$ | $27.99 \pm 0.28$ | $90.84 \pm 0.024$ |
| T | $15.25 \pm 0.09$ | $-3.37 \pm 0.13$ | $43.80 \pm 0.17$ | $0.329 \pm 0.001$ |

**Table 3** Characterization of the accumulation events. Columns refer to date, water filled pore space (WFPS), observed N$_2$O fluxes ($f$(N$_2$O)$_{GC-ECD}$), Keeling plot-derived SP values, obtained net isotope effect for $\Delta\delta^{15}$N(NO$_3^-$–N$_2$O), obtained net isotope effect for $\Delta\delta^{18}$O(N$_2$O/H$_2$O), fraction of remaining N$_2$O after N$_2$O reduction (rN$_2$O. sc11 = SP vs. $\Delta\delta^{15}$N(NO$_3^-$–N$_2$O) approach scenario 1, sc12 = SP vs. $\Delta\delta^{15}$N(NO$_3^-$–N$_2$O) approach scenario 2, sc21 = SP vs. $\Delta\delta^{18}$O(N$_2$O/H$_2$O) approach scenario 1, and sc22 = SP vs. $\Delta\delta^{18}$O(N$_2$O/H$_2$O) approach scenario 2). Results are sorted by descending WFPS.

| Event number | WFPS (%) | Date (2016) (dd.mmm.) | $f$(N$_2$O)$_{GC-ECD}$ (µg N m$^{-2}$ h$^{-1}$) | SP$_{Keeling}$ (‰) | $\Delta\delta^{15}$N(NO$_3^-$-N$_2$O) (‰) | $\Delta\delta^{18}$O(N$_2$O/H$_2$O) (‰) | rNit1 (a.u.) | rNit2 (a.u.) | rN$_2$O_sc11 (a.u.) | rN$_2$O_sc12 (a.u.) | rN$_2$O_sc21 (a.u.) | rN$_2$O_sc22 (a.u.) |
|---|---|---|---|---|---|---|---|---|---|---|---|---|
| 1 | 98.2 ± 0.2 | 22.Jun. | 50 | 31.9 ± 1.9 | 5.3 ± 6.8 | 74.2 ± 2.6 | 0.38 | 0.43 | < 0.01 | 0.03 | < 0.01 | 0.13 |
| 2 | 97.2 ± 0.6 | 23. Jun. | 147 | 36.7 ± 2.0 | 14.4 ± 4.6 | 73.0 ± 2.3 | 0.50 | 0.58 | < 0.01 | 0.04 | < 0.01 | 0.18 |
| 3 | 95.3 ± 0.9 | 24. Jun. | 318 | 22.4 ± 4.2 | 11.3 ± 5.3 | 51.3 ± 7.0 | 0.36 | 0.36 | 0.04 | 0.10 | 0.14 | 0.41 |
| 4 | 93.2 ± 0.8 | 17. Jul. | 259 | 8.4 ± 3.7 | 24.6 ± 6.3 | 65.6 ± 3.4 | 0.35 | 0.25 | 0.76 | 0.81 | 0.11 | 0.21 |
| 5 | 90.2 ± 1.2 | 18. Jul. | 260 | 34.3 ± 5.2 | 18.7 ± 6.3 | 70.7 ± 2.1 | 0.51 | 0.53 | < 0.01 | 0.07 | < 0.01 | 0.19 |
| 6 | 87.7 ± 12.6 | 12. Jul. | 236 | 13.0 ± 4.5 | 25.9 ± 4.1 | 46.6 ± 3.4 | 0.37 | 0.14 | 0.51 | 0.64 | 0.26 | 0.36 |
| 7 | 85.5 ± 2.7 | 19. Jul. | 560 | 25.7 ± 0.8 | 16.9 ± 4.6 | 74.5 ± 2.7 | 0.41 | 0.24 | 0.02 | 0.12 | 0.02 | 0.01 |
| 8 | 73.8 ± 9.3 | 20. Jul. | 411 | 16.7 ± 2.0 | 18.7 ± 5.6 | 56.7 ± 2.5 | 0.36 | 0.15 | 0.16 | 0.28 | 0.12 | 0.21 |
| 9 | 70.3 ± 1.6 | 09. Jul. | 596 | 29.6 ± 1.0 | 11.5 ± 5.0 | 70.7 ± 3.6 | 0.40 | 0.39 | 0.01 | 0.06 | 0.01 | 0.15 |
| 10 | 66.0 ± 1.9 | 21. Jul. | 475 | 33.2 ± 7.9 | 21.1 ± 3.9 | 62.3 ± 6.6 | 0.53 | 0.58 | < 0.01 | 0.10 | < 0.01 | 0.32 |
| 11 | 64.1 ± 2.1 | 10. Jul. | 340 | 33.0 ± 3.9 | 10.1 ± 5.0 | 60.0 ± 2.0 | 0.42 | 0.60 | < 0.01 | 0.04 | < 0.01 | 0.38 |
| 12 | 58.8 ± 1.3 | 11. Jul. | 251 | 15.7 ± 0.7 | 29.5 ± 4.8 | 55.8 ± 1.1 | 0.41 | 0.13 | 0.53 | 0.70 | 0.14 | 0.21 |

**Table 4 Characterization of the lower and upper range for SP, $\Delta\delta\ ^{15}$N and $\Delta\delta\ ^{18}$O for the two N$_2$O-emitting domains fungal denitrification/ nitrification (FD/N) and bacterial denitrification (nitrifier denitrification (BD/ND) according to literature. All values are given in per mil (‰).**

| Parameter | FD/N | BD/ND | Literature |
|---|---|---|---|
| SP | 29.8[i] – 34.5[i] | -5.0[ii] – 3.2[ii] | i) Denk et al. (2017); ii) Lewicka-Szczebak et al. (2017) |
| $\Delta\delta\ ^{15}$N | 30.9[iii] – 68.0[iv] | 0.0[iv] – 37 [iv] | iii) Rohe et al. (2014) and iv) Koba et al. (2009) |
| $\Delta\delta\ ^{18}$O | 35.6[v] – 55.2 [v] | 17.4 [v] – 26.5 [v] | v) Lewicka-Szczebak et al. (2017) |

**Notes:** i) 29.8 and 34.5 refer to the 0.25 and 0.75 quantiles of all values compiled by Denk et al. (2017) in Table S12 for the indices 1 – 3. iii) Lowest absolute isotope effect ($\eta$) of NO$_3^-$ reduction to N$_2$O by fungal denitrification as found by Rohe et al. (2014). iv) Taken from Koba et al. (2009) (referring to Yoshida (1988))

## 7.3 List of figure captions

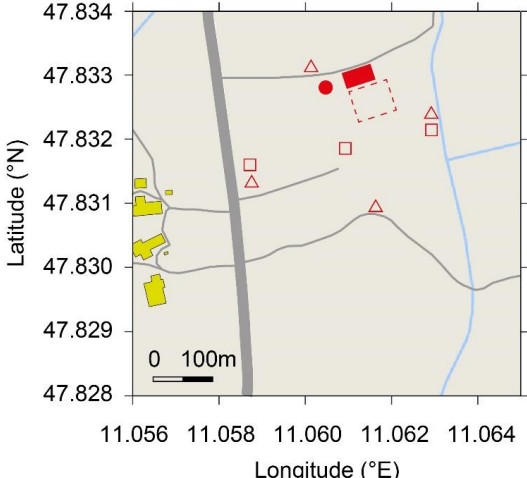

**Figure 1 Map of research site De-Fen with nearby farm buildings (yellow), streets and country lanes (grey) , ditches (blue), the mobile laboratory (filled red square), 2-m sample inlet for TREX-QCLAS measurements, (red dot), area of flux chambers and soil water content measurements (dashed red square), precipitation gauges (open triangles) and location of PT-100 sensors (open squares).**

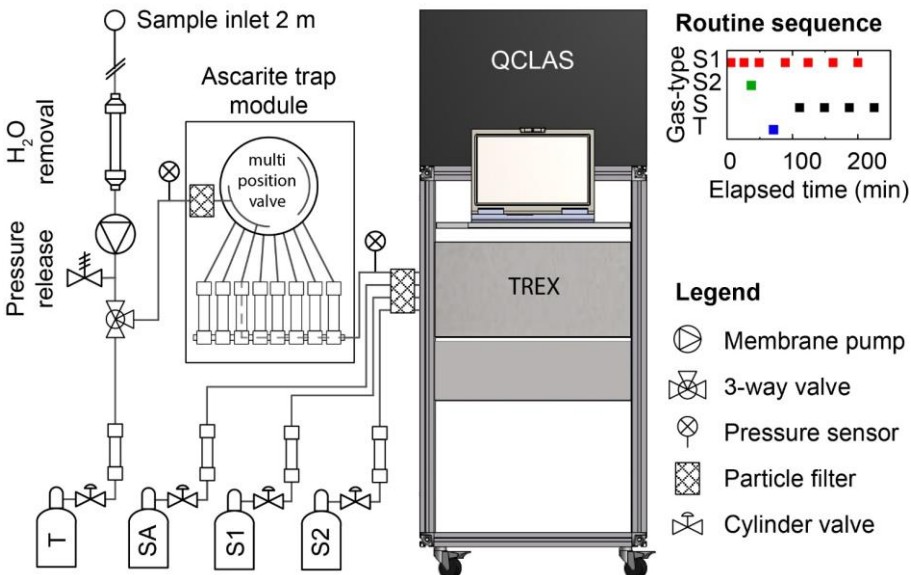

**Figure 2 Instrumental setup for semi-continuous analysis of N₂O isotopes by coupled preconcentration laser spectroscopy (TREX-QCLAS) (Ibraim et al., 2018), including peripherals for the conditioning of the sample gas. Consecutive sample gas treatments include dehumidification by permeation drying, adjustment of sample gas pressure with a pressure release valve after the membrane pump, and $CO_2$ / $H_2O$ removal using Ascarite / $Mg(ClO_4)_2$ traps and filtering for particles using a sintered metal filter. An automized multiposition valve enables us to switch between eight different Ascarite traps and thus reduces the maintenance effort to one visit per eight days. The indicated gases are: target gas (T), synthetic air (SA), standard gas 1 (S1) and standard gas 2 (S2). CO is removed from the analyte gases using a Sofnocat catalyst (type 423, Molecular Products LTD). At the top right, a full measurement cycle is given. Letters on the y axis correspond to different gas types: standard 1 (S1), standard 2 (S2) sample (S) and target (T) gases. The x axis gives the elapsed time in minutes. The full measurement cycle lasts approx. four hours, which results in a frequency of approx. 1 $hr^{-1}$ for ambient air measurements.**

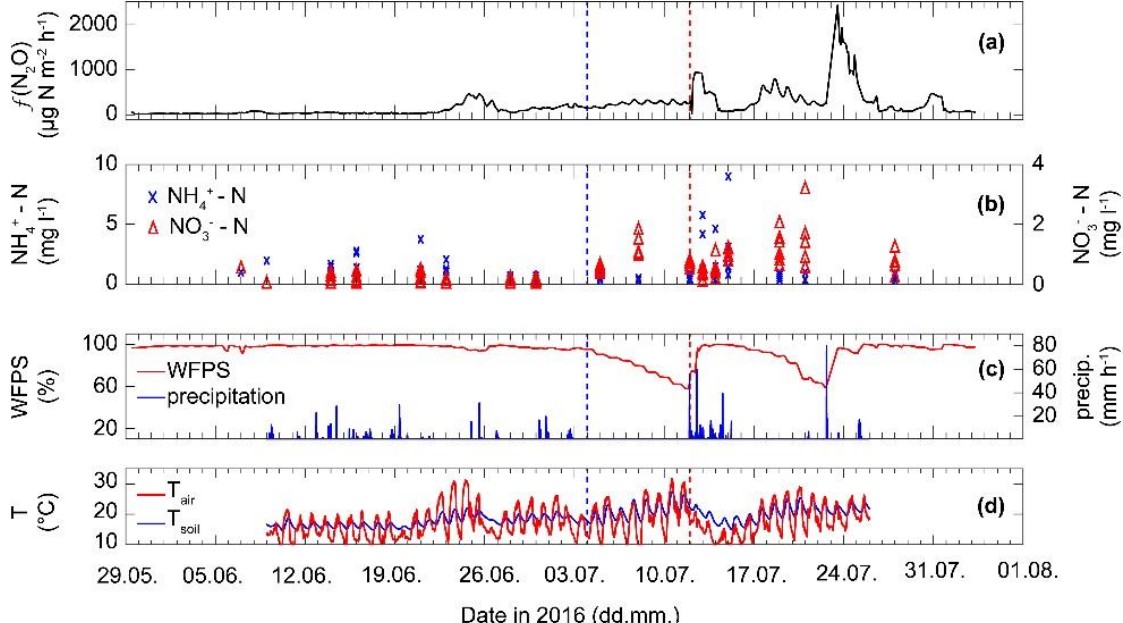

**Figure 3 (a) Average N₂O flux ($f$(N₂O)) as measured by five replicate flux chambers coupled to a GC-ECD. (b) Concentration of NH₄⁺ - N and NO₃⁻ - N. Up to 8 nodes across the De-Fen site were sampled twice per week, except in the week of fertilization, when sampling frequency was increased. Therefore, variability within a sampled day refers to spatial variability across the De-Fen site on the given day. (c) Observed WFPS (red) and precipitation (blue). (d) Observed ambient (2 m above ground) and soil (2 - 6 cm below ground) temperature. The blue dashed line indicates a cutting event and the red line indicates a manure application.**

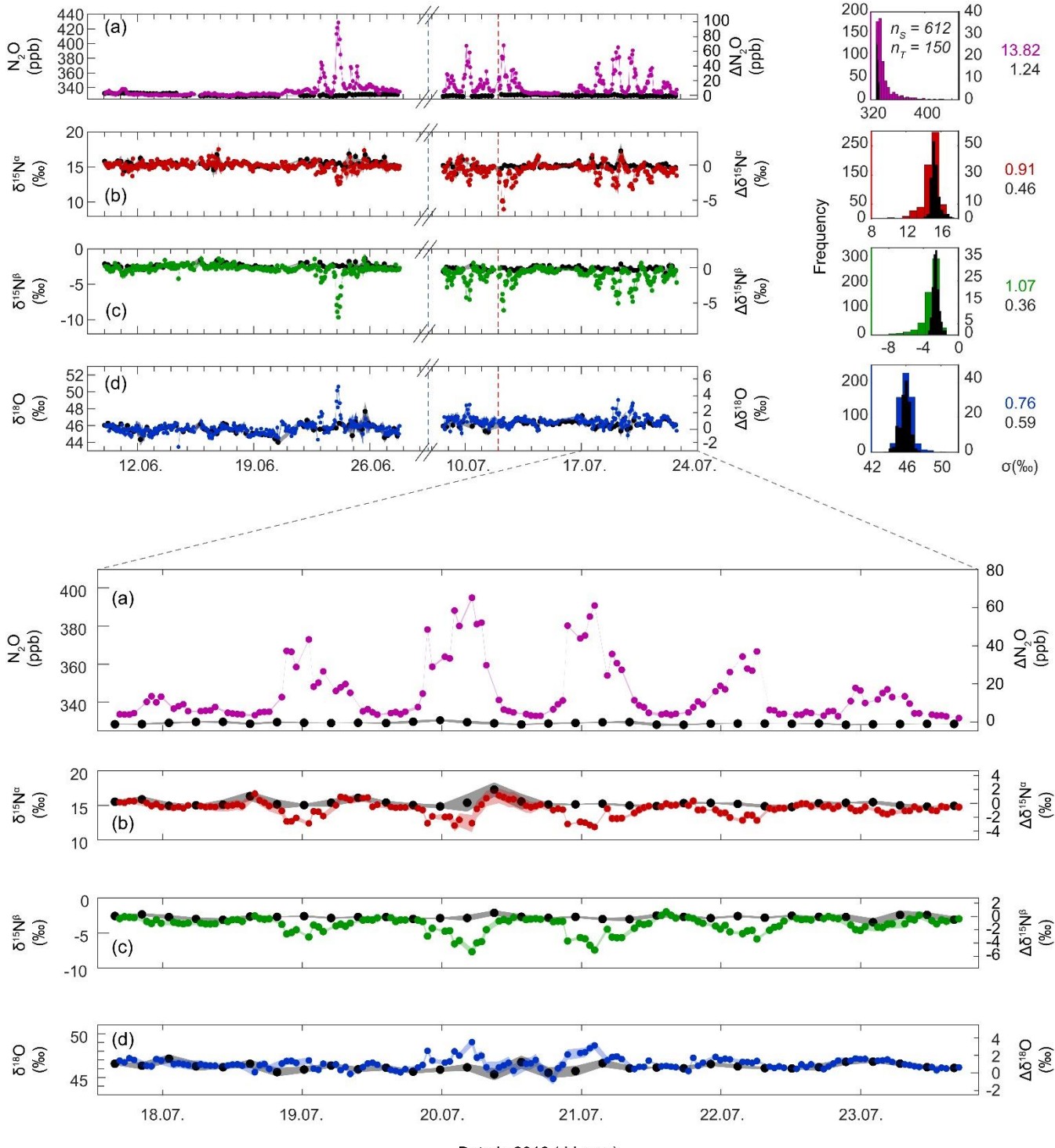

**Figure 4** Time series of N₂O concentrations (a), $\delta^{15}N^{\alpha}$ (b), $\delta^{15}N^{\beta}$ (c) and $\delta^{18}O$ (d), respectively. The left axes give concentrations and isotope delta values on the respective scales, while the right axes depict the difference to background values ($\Delta X = X_{measured} - X_{background}$, where X refers to N₂O, $\delta^{15}N^{\alpha}$, $\delta^{15}N^{\beta}$ or $\delta^{18}O$, respectively). At the top right, histogram plots of the four quantities are given. Coloured symbols indicate ambient air samples (S) from 2 m above ground, and black symbols refer to the corresponding measurements of the target gas (T, Table 2). Shaded areas indicate one standard deviation ($\sigma$) as calculated for three consecutive measurements of T. Standard deviations for the complete measurement period are given on the right, in coloured font for S and in black for T. The vertical blue dashed lines indicates a cutting event on 04 July 2016, while the red dashed line indicates a manure application on 12 July 2016.

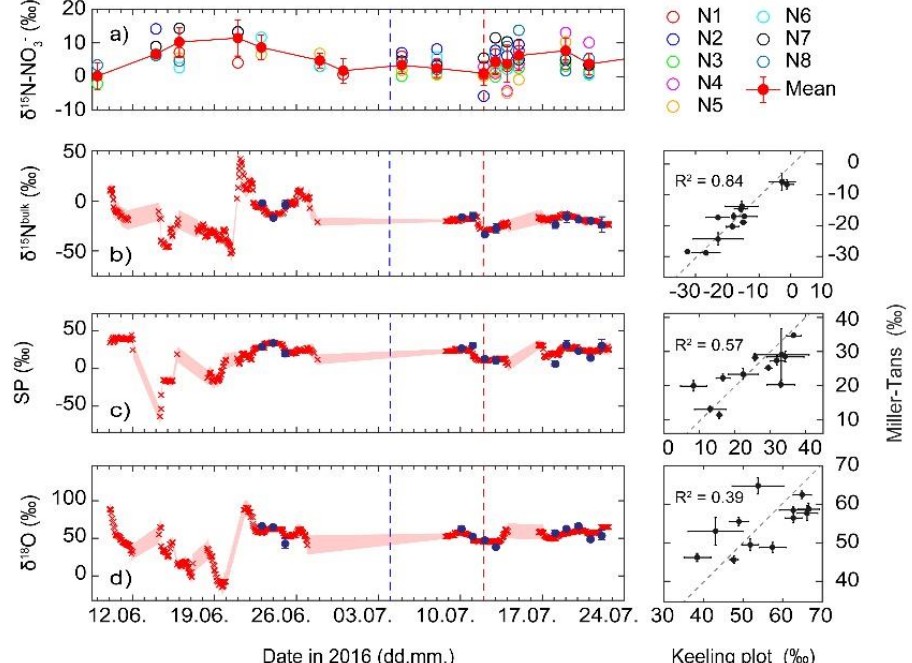

**Figure 5 Temporal trend of δ¹⁵N of soil-extracted NO₃⁻ at eight different nodes (N1-N8) near the N₂O flux and isotope measurements at De-Fen, indicating large spatial variability. In (a), concentration-weighted average values (red filled symbols) and their uncertainty (one standard deviation) are given. Source signatures (b) δ¹⁵Nᵇᵘˡᵏ, (c) SP and (d) δ¹⁸O of soil-emitted N₂O derived from**
5  **the Miller and Tans (2003) approach (red crosses) and the Keeling (1961, 1958) plot approach (blue filled symbols) are given. Uncertainties are indicated as pale red shaded areas for the Miller-Tans approach and error bars for the Keeling plot approach (one standard deviation with a Monte Carlo model). The blue dashed line shows the cutting event, while the red dashed line indicates the manure application. Three panels on the right: correlation diagram of results derived from the Miller and Tans (2003) and the Keeling (1961, 1958) plot approaches. The dashed line corresponds to the 1:1 slope.**

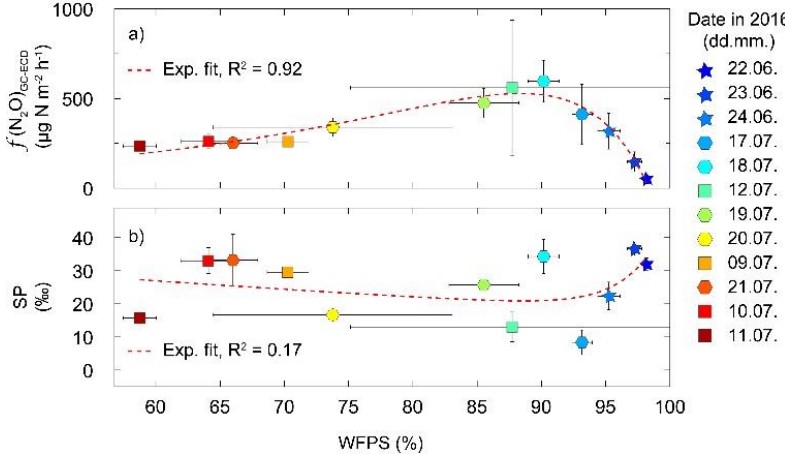

**Figure 6 (a) Noon-to-noon average N₂O flux rates $f$(N₂O) versus water filled pore space (WFPS) from the grassland site De-Fen. Indicated uncertainties represent variations of $f$(N₂O) and WFPS within one day. Results for individual chambers are given in SI Figure 1. N₂O fluxes were highest at 85 – 92 % WFPS. The red dashed line corresponds to a two-term exponential fit of the data shown here. (b) SP as a function of WFPS. Lowest SP values were found in the range of 85 – 95 % WFPS, which corresponds to the**
15  **highest N₂O fluxes. The red dashed line depicts a double exponential fit of the data shown here (this fit was not significant). The model used to fit the data corresponds to y = $a$·exp($b$·x) + $c$·exp($d$·x) (coefficients $a$, $b$, $c$ and $d$ are given in the main text).**

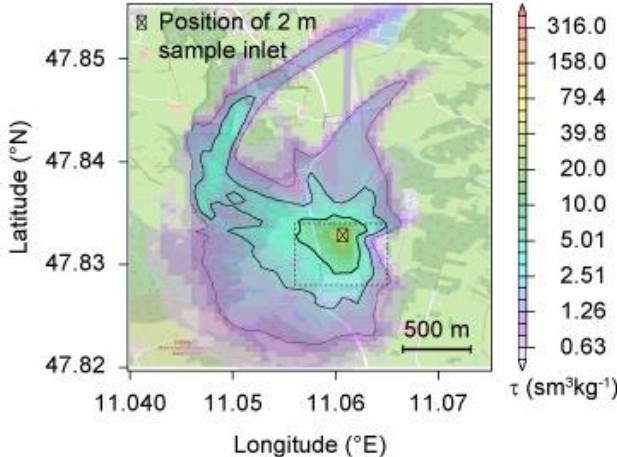

**Figure 7 The average footprint area as calculated by the FLEXPART-COSMO model. The model was driven with local wind vectors and observed N₂O flux rates, and indicates a major contribution of soils south-west of the ambient air inlet. The source sensitivities Tau (τ) are given as the product of residence time (in seconds) and inverse atmospheric density (in m³*kg⁻¹). Isolines enclose the areas of largest source sensitivities summing up to 15, 30 and 45 % (decreasing thickness of lines) of the total simulated source sensitivity. The dashed rectangle indicates the area depicted in Figure 1. Individual source signatures for the 12 events are provided in the supplements (SI Figure 11).**

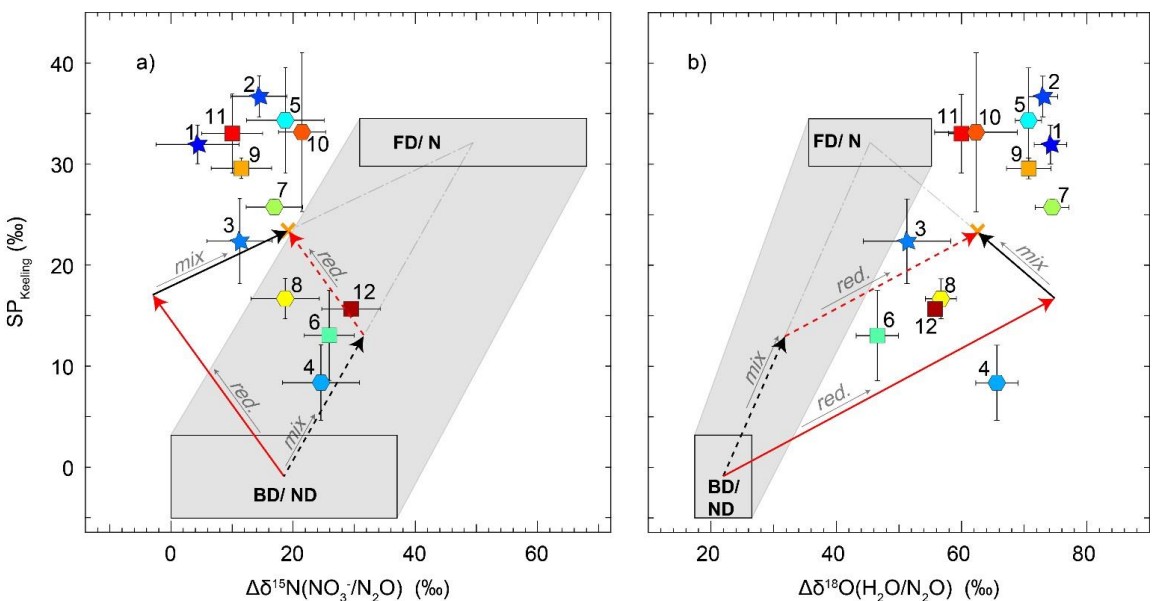

**Figure 8 Source signatures of soil-emitted N₂O for 22 to 24 June (stars 1-3), July 9 to 12 (squares 6, 9, 11 and 12) and July 17 to 21 (hexagons 4, 5, 7, 8 and 10). The colour coding of the symbols refers to WFPS, where blue corresponds to high and red corresponds to low WFPS (exact values in Table 3). The source signatures of fungal denitrification- and nitrification-derived N₂O (FD/N) and bacterial denitrification- and nitrifier denitrification-derived N₂O (BD/ND) are highlighted with rectangles according to the values given in Table 4, and the shaded area represents the mixing region of the two domains. The orange cross indicates the flux averaged values of the respective source signatures. Red arrows denote the path of partial N₂O reduction to N₂, while black arrows indicate the direction of mixing with FD/N-derived N₂O. Solid arrows indicate scenario 1 (first reduction, then mixing), while dashed arrows indicate scenario 2 (first mixing, then reduction). (a) SP versus Δδ¹⁵N map according to Koba et al. (2009), where $\Delta\delta^{15}N = \delta^{15}N\text{-}NO_3^- - \delta^{15}N\text{-}N_2O$ (b) SP versus Δδ¹⁸O of soil-emitted N₂O according to Lewicka-Szczebak et al. (2017), where $\Delta\delta^{18}O = \delta^{18}O\text{-}N_2O - \delta^{18}O\text{-}H_2O$, the isotope effect between soil water and formed N₂O. Δδ¹⁸O for BD/ND is considered to be constant, because of the high oxygen exchange between soil water and reaction intermediates at high WFPS. An exemplary illustration is provided in the supplementary (SI Figure 4).**

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
