# Peer review of "Attribution of N2O sources in a grassland soil with laser spectroscopy based isotopocule analysis"

_Biogeosciences, 2018_

## Short Comment (SC1) · 21 Nov 2018

A very different method to determine N2O emission fluxes was applied in Schäfer et al. (2012) to determine such fluxes from unfertilized grassland on the field scale. Fluxes in about the same amount, which are found and described here by Ibraim et al. in chapter 3.1 "N2O fluxes and soil parameters", were determined in our investigations by a tunnel, coupled to an open-path Fourier transform infrared spectrometer, which covered 500 m2, from 0 up to 14 $\mu$g N2O-N m−2 h−1. The FLEXPART-COSMO simulations had source sensitivity which originates from areas within approximately 300 m to 700 m distance to the sample inlet representing an up-scaling similar to the

tunnel method (100 m length). Peak emissions were detected in Schäfer et al. (2012) by concurrent chamber measurements after rainfall as described by Ibraim et al. also. Schäfer, K., Böttcher, J., Weymann, D., Von der Heide, C., Duijnisveld, W.: Evaluation of a closed tunnel for field-scale measurements of N2O fluxes at the soil-atmosphere interface. J. Environ. Qual., 41, 1383-1392 (2012); doi: 10.2134/jeq2011.0475.

––––––––––––––––––––––––––––

---

## Short Comment (SC2) · 26 Nov 2018

We appreciate the comment of Klaus Schäfer and agree that both approaches, tunnel / FTIR and TREX-QCLAS / FLEXPART-COSMO, can be applied to quantify areal N2O emissions from grassland sites. In addition, a second technique, coupled flux-chambers and GC-ECD, were used in our study, and average N2O fluxes determined with five chambers agree with TREX-QCLAS / FLEXPART-COSMO (and tunnel / FTIR) results (SI Fig. 2). To acknowledge the technique used by Klaus Schäfer and co-authors, we will refer to their work in our revised manuscript.

---

## Short Comment (SC3) · 26 Nov 2018

We thank you very much for this comprehensive overview of available techniques and discussion. I wish you all the best for your further research in this field where you are on the right way.

---

## Referee Comment (RC1) · Ostrom (Referee) · 30 Nov 2018

Review of Ibraim et al
        Nathaniel Ostrom, Michigan State University

        The authors present an innovative and thorough assessment of the microbial origins of $N_2O$ emissions from a grassland soil based on spectroscopic measurements.  This data set is clearly an advance for the field as extensive time-series data sets for the isotopic composition of $N_2O$ are rare but very important to constrain the dynamic nature of $N_2O$ production from soils. While I appreciate the thoroughness of the data set and interpretations I have a few central concerns regarding calibration and with the assessment of microbial origins of $N_2O$ described in the paper.
        Characterization of sample isotope values based on two isotopically characterized standards is certainly the minimum necessary. Critical is also that the range of isotope values of the samples be encompassed by the range in isotope values of the standards. The range in isotope values for $\delta^{15}N^{\alpha}$ and $\delta^{15}N^{\beta}$ is quite good (although I don't know what fraction of the samples lie outside these ranges) the range in $\delta^{18}O$ values of the standards is small and does not well encompass the range in sample values. The issue with this is not a just with the precision (which can be determined) but with the accuracy; values outside the range of the standards cannot be considered accurate even if precise. My personal opinion is to not publish isotope values outside the range of standards however, I appreciate that there may be a statistical approach for providing greater confidence in this situation.
        While the authors present an extensive data set on the concentration and isotopic composition of $N_2O$ in air much of the data set is derived from periods of low flux when the concentration of $N_2O$ is only slightly above atmospheric. The calculation of soil-derived $N_2O$ is obtained from a simple isotope mixing model (soil-derived mixing with atmospheric $N_2O$). In this model, there is considerable error associated with the soil-derived isotope values at low $N_2O$ concentration (the error increases as the concentration of $N_2O$ declines). Further, the error around each calculated soil-derived isotope value increases as the difference in the isotopic composition between soil and atmospheric $N_2O$ decreases. For this reason, in our laboratory, we generally don't publish data on samples in which the $N_2O$ concentration is less than 30% greater than atmospheric (although this is an arbitrary value). The Keeling-plot approach does provide greater accuracy and precision but it is critical that the authors (1) report the uncertainty obtained for all soil-derived isotope values, (2) provide a clear discussion of this issue, and (3) consider not reporting isotope values for which the error is very large and a reasonable explanation for what the cut-off error should be.
        There are two important assumptions in the graphical approach to tracing the microbial origins of $N_2O$ that is used by the authors: (1) that $\delta^{18}O$ is a conservative tracer of origins and (2) that the kinetic isotope effects associated with production of $N_2O$ as well as reduction of $N_2O$ are constants. As I discuss below I question the validity of both of these assumptions. I will acknowledge, however, that this is a common approach in the literature but ask that the authors, at least, discuss

these issues and how variation in the kinetic isotope effects might impact their model outcomes.

Page 1, Line 17: Was data corrected based on measured values for the Target gas?

Page 5, Line 26: Was the concentration correction performed daily? Did the range in concentration standards encompass the range in sample concentrations observed?

Page 5, line 34: Did the Keeling plot approach include calculation of the uncertainty surrounding the isotope values for the soil derived $N_2O$? Was this evaluated using the Monte Carlo model as described? At concentrations only slightly above ambient this error can be very large. We generally discard data with a $N_2O$ concentration < 30% above atmospheric levels although the decision to discard data should probably be made on the basis of the magnitude of the uncertainty.

Page 5, Line 36: Why was a correlation between $\delta^{15}N^{\alpha}$ and $\delta^{15}N^{\beta}$ considered a criterion for a valid measurement? While unpublished, our data indicates that most of the variation in SP during $N_2O$ production from the cNOR enzyme is driven by $\delta^{15}N^{\beta}$; which suggests that a lack of correlation could be a normal feature of production from denitrification.

Page 6, Line 5: It is not really necessary to specify the "hard disk" failure as the cause of missing data. Simply stating equipment failure is fine. This is also mentioned in the Figure 4 legend.

 Page 7, line 26: Delete "during": word not needed.

Page 8, Line 22: On what basis was an $r^2$ of 0.2 similarly used to exclude data? This seems like a very poor degree of correlation.

Page 8, Line 23: Perhaps "calculated" is a better word than "extracted".

Page 8, line 24: What are "relatively large uncertainties"? It would be better to state the uncertainty in the text as well as in the figures. Perhaps it would be best to exclude data with large uncertainties; as long as some reasonable criteria can be established to exclude data beyond a certain uncertainty (perhaps based on the range in isotope values between the sources).

Page 8, line 30: Yes, the poorer data quality for $\delta^{18}O$ is likely a consequence of the fact that the standards do not encompass the data. Generally, standards must encompass the range in data to assure accuracy in any calibration relationship. The issue is that this is does not only result in poorer precision but, more importantly, results in poorer accuracy. The authors are presenting data for which the accuracy

is questionable. If the $\delta^{18}O$ values obtained are substantially outside the range of the standards the authors should give considerable thought as to whether they should be excluded.

Page 9, line 19: I am not fond of the graphical approach used by Koba et al to apportion sources of $N_2O$ largely because $\delta^{18}O$ is a poor tracer of microbial origins. It is well known that oxygen exchanges with water during the microbial production of $N_2O$ which can alter $\delta^{18}O$ values (e.g. Kool references). Further, this approach relies on limited data sets for $\delta^{18}O$ generated by cultures of fungal denitrification, bacterial denitrification and nitrification.  Given this, I suggest that the authors, at least, acknowledge the limitations of the graphical approach in assessment of the origins of $N_2O$.

   For $\delta^{15}N$, this approach also depends not only on the isotopic composition of the nutrient source but also the kinetic isotope effect (KIE) associated with $N_2O$ production. The KIE is more accurately described as a net isotope effect and can be highly variable depending upon conditions in the soil. The authors need to consider and discuss the effect of variation in the KIE on their model outcomes.

Page 10, Line 5: Here again a single value is chosen for the KIE associated with $N_2O$ reduction whereas there are many papers that show variation. In Jinuntuya-Nortman et al (2010) we demonstrate that the SP net isotope effect associated with $N_2O$ reduction can approach zero per mil and is inversely correlated with water filled pore space. The authors demonstrate that water filled pore space varies. What is the affect of a varying NIE on the model outcomes? Indeed their inference that low SP values occur at high WFPS is entirely consistent with the role of WFPS in masking expression of fractionation during $N_2O$ reduction and may not reflect variation in the proportion of $N_2O$ derived from nitrification and denitrification. The estimates of the rates of $N_2O$ reduction may not be meaningful if the net isotope effect for reduction varies (as it likely does).

Page 12, line 36: Given the issues raised above I am not convinced by the author's statement that natural abundance isotope studies are "an effective way to disentangle $N_2O$ production pathways". A frank assessment of the strengths and weaknesses of the approach is needed.

---

## Referee Comment (RC2) · Anonymous Referee #2 · 8 Jan 2019

Review of Ibraim et al 2018 The paper by Ibraim and co-authors presents a novel technique to measure a suite of isotopic fingerprints of N2O using a field-deployable device. Since N2O emissions and their isotopic ratios vary spatially and temporally to a large degree, such an instrument is very useful to allow extent our knowledge on the processes driving these N2O emissions. The measurements presented in the paper demonstrate that the QCLAS instrumentation works well under field conditions and allows its implementation in further studies. This successful field deployment is certainly a central step after a year-long construction and test phase in the laboratory and I congratulate the team for this effort. Likewise it is clear to the authors of this study that measuring the isotopic information of N2O is just one step, and to interpret

and exploit the data requires extensive other knowledge, ranging from meteorological boundary conditions, soil analyses, to interpreting the application of manure on the sampling site. As further outlined below, some of these non-measurement aspects of the paper should be improved to gain more clarity for the readers. Since the current measurement set up was not perfect, i.e. only night-time measurements provide robust results, a lot can be learned from this measurement campaign for future field applications. At some places the authors already suggested how these limitations could be overcome with a better design etc. I have the impression that these "lesson learned statements" could be extended to guide future measurement campaigns in this area. In general, the paper is clearly structured, the figures are mostly instructive and the text is written nicely. I therefore welcome this paper for final publication after my and the other reviewers comments are included in the final version.

General observations and detailed comments: In section 4.4.6 (page 13) you discuss the influence of the manure application (12th July) on the calculated source signatures of the emitted N2O. A causal link between excessive nitrogen addition on subsequent N2O emissions from the soil is to be expected and might be the case. However, looking at the flux time series in Fig. 3 it is equally clear that N2O fluxes rise after intense rainfall events and this also fulfils expectation. The highest N2O emissions within the entire study follow the strong rainfall event end of July - here without a manure application. However, the manure application on the 12th July is almost synchronous with the rainfall event; the farmer apparently waited for rain to apply the manure. My feeling is that the discussion in section 4.4.6 focusses too strongly on the manure application, while the more likely driver behind the rising N2O emissions (intense rainfall) is not really discussed equally. A second argument for the "heavy rain hypothesis" is also the wide footprint of the isotope measurements. From Fig. 7 I get the impression that the largest fraction of the emissions stems from outside of the dashed rectangle where the site is located. Only 15 to 30% of the N2O emissions came from the local field where the manure was applied, while the largest fraction comes from an area of a few km distance. It might be that the other fields in the surroundings were also fertilized at the

same time by the farmers and thus the De-Fen site is representative, but this information is missing. I might be wrong, but you might gain additional insight to view the N2O flux data and the isotopic signatures also from this heavy rainfall point of view (likewise WFPS) and a wider, more realistic regional footprint. In this respect you might also put less weight on the NH4 and NO3 soil extracted solute data because these data might be too local for the footprint of the measured N2O isotope signatures. As suggested in your conclusions, it will be valuable in future measurement campaigns to also sample air from chambers that are more representative of the site. In other words, the precious isotope measurement time during the day or for meteorological situations that do not lead to sufficient N2O accumulation in the boundary layer could be better invested. Please mention the footprint shown in Fig. 7 earlier in the paper. It would be helpful for some readers (including me) to be aware that the actual footprint of the N2O isotope data is more extended than what is visualized in Fig. 1.

Page 1, Lines 33 and 37: Mentioning the sink term "while N2O reduction acted as a major sink" may not clear to all readers. Does this refer to a consumption of N2O produced in the soil itself or also for ambient atmospheric N2O, i.e. a net sink to the atmosphere?. "N2O reduction to N2 largely dictated the isotopic composition of measured N2O. " Does this statement refer to all measured isotope ratios; this statement seems very general.

Page 3, Lines 35: stick to one name for the management "cutting" vs moving (caption Fig. 4) Page 3, Line 30: Site name: Could you use just Fendt as the site name in your paper rather than the awkward De-Fen (I know the acronym De-Fen is the more official in terms of the European Flux Database cluster).

Page 3, Lines 37: I am not a specialist for agricultural manures, but my understanding is that manure usually refers to animal feces (with the N mostly in form of urea) so I am confused by the ammonium N and referring to the Raiffeisen Laborservice; Do you mean inorganic fertiliser e.g. pure Ammonium sulfate? I any case please specify this. Further, I wonder if it would have been worthwhile to obtain also the bulk N-isotopic

composition of the two different kinds of fertilizers/manure. You put a lot of effort into measuring the spatial and temporal distribution of the ïĄď15N of soil-extracted nitrate while a value for the manure might be valuable as well for the input signature of d15N of soil NH4.

Page 4, Line 14: This measurement-specific information seems not necessary here ("While...) and could be deleted.

Page 5, Line 2: sentence could be shortened: "Then the gas was dried using a Nafion dryer...()....also, delete: overpressured (the 4.5 bar already indicate that) Page 5, Line 36: Given the complexity of the pathways, this correlation criterion is not a sound argument for a valid measurement as it discards 18 out of 30 values while accepting 12 only leading to a bias in the results.

Page 8, Line 32. You could end the sentence after the Toyoda citation and delete after ", but..." as this does not add much.

On page 8, line 35 you write: "At night, within a stable nocturnal boundary layer, vertical wind speeds and hence tracer transport are low, while lateral wind speeds can be high and constituents like N2O can be transported over larger distances. As a result, N2O emissions from other land uses or land cover may have contributed to the observed N2O isotopic composition. To assess the possible influence of other land use / land cover. Please omit may and possible in these occasions where you actually know that more distant emission contribute.

Table 1: unit for bulk density is not % but rather g/cm3; pH is dimensionless; Table: 2: to prevent confusion with the units, provide all values for the mole fraction in ppm, i.e. for T 0.329 ppm. Table 3: Event no (a.u)? did not get that for the three columns with SPKeeling and ïĄď15N and ïĄď18O: one digit seems enough for the +- values, i.e. 1.9 instead of 1.91 for SP. Table 4: caption: better: Characterization of the lower and upper range for... first column header "Source signature" should read parameter or signature

Figure 3: Note, the x axis label for Fig. 3 and 4 and 5, 6 are all different (Date vs Datum in). Please select one for all, e.g. Date in 2016. panel c with precipitation: 80 mm per hour seems a very high value, please check. please rewrite sentence to omit, respectively: "Blue and red dashed lines refer to a cutting event and to a manure application, respectively." to: The blue dashed line indicates a cutting event and the red line manure application. (similar as you write in caption Fig. 4).

Figure 4: please explain the values given on the right side of the histograms zoom panel a: at around 20.7. there is a weird magenta dot within the background values (black dot) caption: please replace y-axis with axes (plural) general observation at Fig. 4: In Fig. 3 it becomes apparent that the heavy precipitation events (around 12.7. and 22.7.) that lead to a progressive reduction of the WFPS are strongly connected with two prominent N2O fluxes. While the first heavy rain event (around 12.7.) is connected with the manure application, the second rainfall event happens without manure application. It would be worthwhile to add these heavy rainfall events also in Fig. 4 with lines or other markers.

Figure 7: If possible adjust the colour legend to rounded numbers rather than 3.16e+02, e.g. 0.5; 1; 5; . . .300. Also, if possible, add the numbers (15, 30, 45 %) of the source sensitivities onto the isolines of figure itself (this is quicker than having this written in the caption.

Please also note the supplement to this comment:
https://www.biogeosciences-discuss.net/bg-2018-426/bg-2018-426-RC2-supplement.pdf

---

## Author Comment (AC2) · 20 May 2019

We would like to thank the anonymous referee #2 for his detailed review of our manuscript and for his precious comments and suggestions, which have led to a clear improvement of our manuscript. Below, the referee's comments (RC2) are displayed followed by the authors' response, which are highlighted in grey.

General observations and detailed comments:

The paper by Ibraim and co-authors presents a novel technique to measure a suite of isotopic fingerprints of $N_2O$ using a field-deployable device. Since $N_2O$-emissions and their isotopic ratios vary spatially and temporally to a large degree, such an instrument is very useful to allow extent our knowledge on the processes driving these $N_2O$ emissions. The measurements presented in the paper demonstrate that the QCLAS instrumentation works well under field conditions and allows its implementation in further studies. This successful field deployment is certainly a central step after a yearlong construction and test phase in the laboratory and I congratulate the team for this effort. Likewise it is clear to the authors of this study that measuring the isotopic information of $N_2O$ is just one step, and to interpret and exploit the data requires extensive other knowledge, ranging from meteorological boundary conditions, soil analyses, to interpreting the application of manure on the sampling site. As further outlined below, some of these non-measurement aspects of the paper should be improved to gain more clarity for the readers. Since the current measurement set up was not perfect, i.e. only nighttime measurements provide robust results, a lot can be learned from this measurement campaign for future field applications. At some places, the authors already suggested how these limitations could be overcome with a better design etc. I have the impression that these "lesson learned statements" could be extended to guide future measurement campaigns in this area. In general, the paper is clearly structured, the figures are mostly instructive and the text is written nicely. I therefore welcome this paper for final publication after my and the other reviewers comments are included in the final version.

RC2 Comment 1

In section 4.4.6 (page 13) you discuss the influence of the manure application (12th July) on the calculated source signatures of the emitted $N_2O$. A causal link between excessive nitrogen addition on subsequent $N_2O$ emissions from the soil is to be expected and might be the case. However, looking at the flux time series in Fig. 3 it is equally clear that $N_2O$ fluxes rise after intense rainfall events and this also fulfils expectation. The highest $N_2O$ emissions within the entire study follow the strong rainfall event end of July - here without a manure application. However, the manure application on the 12th July is almost synchronous with the rainfall event; the farmer apparently waited for rain to apply the manure. My feeling is that the discussion in section 4.4.6 focusses too strongly on the manure application, while the more likely driver behind the rising $N_2O$ emissions (intense rainfall) is not really discussed equally. A second argument for the "heavy rain hypothesis" is also the wide footprint of the isotope measurements. From Fig. 7 I get the impression that the largest fraction of the emissions stems from outside of the dashed rectangle where the site is located. Only 15 to 30% of the $N_2O$ emissions came from the local field where the manure was applied, while the largest fraction comes from an area of a few km distance. It might be that the other fields in the surroundings were also fertilized at the same time by the farmers and thus the De-Fen site is representative, but this information is missing. I might be wrong, but you might gain additional insight to view the $N_2O$ flux data and the isotopic signatures also from this heavy rainfall point of view (likewise WFPS) and a wider, more realistic regional footprint. In this respect, you might also put less weight on the $NH_4$ and $NO_3$ soil extracted solute data because these data might be too local for the footprint of the measured $N_2O$ isotope signatures.

**Authors' response:** The authors agree that $N_2O$ emissions and isotopic signatures are driven by both manure (substrate) addition and rainfalls (WFPS) and it might not be possible to disentangle both effects for the manure application on 12 July. The effect of WFPS is explicitly discussed in the section 4.1 ("$N_2O$ fluxes and WFPS") as well as in the sections 4.4.3 and 4.4.5. Nevertheless, we acknowledge the reviewer's comment on the fact that the $N_2O$ emissions strongly raised after rainfall events since each of the three major rainfall events on 24 June, 12 July and 23 July was followed by a clear increase of the $N_2O$ emission rates, which is in agreement with literature (Peng et al., 2011; Liu et al., 2014). Unfortunately, for the two "rainfall – only" events no $N_2O$ source signatures could be retrieved: around 24[th] June the observed data did not allow for significant Keeling plot analysis, while on 23[rd] July the measurement campaign was terminated and the fractional measurement data did not allow calculation of $N_2O$ isotopic composition. We also agree that the footprint of the $N_2O$ isotope measurements goes beyond the fertilized plot. For this reason, we added a number of statements to section 4.4.6:

Page 13 line 7: After the manure application on 12 July and rainfall events in the days thereafter a strong …

Page 13 Line 9: … in the isotopic composition of the applied precursors, by an enhanced fractionation due to higher substrate availability or changes in process conditions (e.g. WFPS, see sections above).

Page 13 Line 14: … during nitrification may have been used as substrate for denitrification, given the increase in WFPS due to intensive rainfall events.

RC2 Comment 2

As suggested in your conclusions, it will be valuable in future measurement campaigns to also sample air from chambers that are more representative of the site. In other words, the precious isotope measurement time during the day or for meteorological situations that do not lead to sufficient $N_2O$ accumulation in the boundary layer could be better invested.

**Authors' response:** In our second field study, which took place between July and December 2017 in Central Switzerland, we deployed automated flux-chambers at the day-time according to a specifically designed measurement schedule. This work is currently in preparation for publication.

RC2 Comment 3

Please mention the footprint shown in Fig. 7 earlier in the paper. It would be helpful for some readers (including me) to be aware that the actual footprint of the $N_2O$ isotope data is more extended than what is visualized in Fig. 1.

**Authors' response:** We agree and we add the following sentence to the end of the section 2.1.1, where the De-Fen site is characterized:

Page 3 line 37: "[…] respectively (Raiffeisen Laborservice, Ormont, Germany). The average footprint area for $N_2O$ flux and isotope measurements is given in Figure 7"

RC2 Comment 4

Page 1, Lines 33 and 37: (i) mentioning the sink term" while $N_2O$ reduction acted as a major sink" may not clear to all readers. Does this refer to a consumption of $N_2O$ produced in the soil itself or also for ambient atmospheric $N_2O$, i.e. a net sink to the atmosphere? (ii) "$N_2O$ reduction

to $N_2$ largely dictated the isotopic composition of measured $N_2O$. " Does this statement refer to all measured isotope ratios; this statement seems very general.

**Authors' response:**

(i)        Page 1 line 37: In the given context our statement referred to soil produced $N_2O$. We cannot exclude reduction of ambient $N_2O$ but at least no net uptake of $N_2O$ was observed. To emphasize this fact we added the following statement: "[…] source for $N_2O$, while $N_2O$ reduction acted as a major sink for soil produced $N_2O$."

(ii)      The dual isotope mapping approach (Fig. 8) indicates that irrespective of the selected approach (SP versus $\Delta\delta^{15}N$ or SP versus $\Delta\delta^{18}O$) and scenario (1: reduction first, 2: mixing first) differences in SP to pure BD/ND were mostly controlled by $N_2O$ reduction. We therefore would like to keep this statement.

RC2 Comment 5

Page 3, Lines 35: stick to one name for the management "cutting" vs moving (caption Fig. 4)

**Authors' response:** Done. "Mowing" was changed to "cutting" at the following positions: page 6 line 34, page 6 line 40 and caption of Figure 4 at page 21 line 9.

RC2 Comment 6

Page 3, Line 30: Site name: Could you use just Fendt as the site name in your paper rather than the awkward De-Fen (I know the acronym De-Fen is the more official in terms of the European Flux Database cluster).

**Authors' response:** We would like to keep "De-Fen" for consistency with other work related to this research site.

RC2 Comment 7

Page 3, Lines 37: (i) I am not a specialist for agricultural manures, but my understanding is that manure usually refers to animal feces (with the N mostly in form of urea) so I am confused by the ammonium N and referring to the Raiffeisen Laborservice; Do you mean inorganic fertiliser e.g. pure Ammonium sulfate? In any case, please specify this. (ii) Further, I wonder if it would have been worthwhile to obtain also the bulk N-isotopic composition of the two different kinds of fertilizers/manure. You put a lot of effort into measuring the spatial and temporal distribution of the $\delta^{15}N$ of soil-extracted nitrate while a value for the manure might be valuable as well for the input signature of $\delta^{15}N$ of soil $NH_4$

**Authors' response:**

(i)        The referee is correct, manure (and not ammonium sulphate) was applied. Raiffeisen Laborservice offers manure analysis (including analysis of N contents). Please see here: https://www.raiffeisen-laborservice.de/biogas/analysen-guelle.

(ii)      We agree that it would have been beneficial to get the manure analysed for the $\delta^{15}N$ content, especially for the interpretation of the event around 12 July. Unfortunately, we did not sample the manure that was applied at De-Fen for subsequent analysis.

RC2 Comment 8

Page 4, Line 14: This measurement-specific information seems not necessary here ("While. . .) and could be deleted.

**Authors' response**: we agree. The sentence "While $NH_4^+$ was converted to an indophenol complex, $NO_3^-$ was reduced to nitrite to produce the diazo chromophore." was deleted.

RC2 Comment 9

Page 5, Line 2: sentence could be shortened: "Then the gas was dried using a Nafion dryer. . .().. . ..also, delete: overpressured (the 4.5 bar already indicate that)

**Authors' response**: The sentence was adapted as follows "Then the sample gas was dried using a nafion drier […]". We would like to keep the term "sample gas" in order to emphasize the gas type. The term "overpressure", however, was deleted.

RC2 Comment 10

Page 5, Line 36: Given the complexity of the pathways, this correlation criterion is not a sound argument for a valid measurement as it discards 18 out of 30 values while accepting 12 only leading to a bias in the results.

**Authors' response:** The main reason for discarding many Keeling plot derived source signatures, e.g. in the period 10 June to 21 June, was the low accumulation of $N_2O$. However, since both reviewers commented on the selection criteria for valid source signatures, we adopted the criteria as stated in our comment 2 to reviewer 1.

RC2 Comment 11

Page 8, Line 32. You could end the sentence after the Toyoda citation and delete after ", but. . ." as this does not add much.

**Authors' response:** We would like to keep this part of the sentence to highlight differences between agricultural and suburban sites.

RC2 Comment 12

On page 8, line 35 you write: "At night, within a stable nocturnal boundary layer, vertical wind speeds and hence tracer transport are low, while lateral wind speeds can be high and constituents like $N_2O$ can be transported over larger distances. As a result, $N_2O$ emissions from other land uses or land cover may have contributed to the observed $N_2O$ isotopic composition. To assess the possible influence of other land use / land cover. Please omit may and possible in these occasions where you actually know that more distant emission contribute.

**Authors' response:** We agree, done.

RC2 Comment 13
  (i)    Table 1: unit for bulk density is not % but rather g/cm3; pH is dimensionless
  (ii)   Table: 2: to prevent confusion with the units, provide all values for the mole fraction in ppm, i.e. for T 0.329 ppm.
  (iii)  Table 3: Event no (a.u)? did not get that for the three columns with SPKeeling and ïAₑd'15N and ïAₑd'18O: one digit seems enough for the +- values, i.e. 1.9 instead of 1.91 for SP.

(iv)   Table 4: caption: better: Characterization of the lower and upper range for. . . first column header "Source signature" should read parameter or signature

**Authors' response:**

(i)   Done
(ii)  Done
(iii) 'Event no. (a.u.)' was changed to 'Event number' and the number of digits was reduced to one as suggested
(iv)  We agree, done

RC2 Comment 14

(i)   Figure 3: Note, the x axis label for Fig. 3 and 4 and 5, 6 are all different (Date vs Datum in). Please select one for all, e.g. Date in 2016.
(ii)  panel c with precipitation: 80 mm per hour seems a very high value, please check.
(iii) please rewrite sentence to omit, respectively: "Blue and red dashed lines refer to a cutting event and to a manure application, respectively." to: The blue dashed line indicates a cutting event and the red line manure application. (similar as you write in caption Fig. 4).

**Authors' response:**

(i)   Done, 'Date in 2016 (dd.mm.)' was chosen for all
(ii)  Done
(iii) Done

RC2 Comment 15

(i)   Figure 4: please explain the values given on the right side of the histograms
(ii)  zoom panel a: at around 20.7. there is a weird magenta dot within the background values (black dot)
(iii) caption: please replace y-axis with axes (plural) general observation at Fig. 4: In Fig. 3 it becomes apparent that the heavy precipitation events (around 12.7. and 22.7.) that lead to a progressive reduction of the WFPS are strongly connected with two prominent $N_2O$ fluxes.
(iv)  While the first heavy rain event (around 12.7.) is connected with the manure application, the second rainfall event happens without manure application. It would be worthwhile to add these heavy rainfall events also in Fig. 4 with lines or other markers.

**Authors' response:  We agree and we will make the following changes:**

(i)   Done
(ii)  Done.
(iii) Done.
(iv)  The precipitation mentioned by the reviewer occurred on 23 July. As the scaling of Fig. 3 and 4 are different, the rainfall event occurs at the end of the $N_2O$ isotope measurements and did not show up in the results. We therefore prefer to not add a vertical line on the last day of this plot, because possible consequences of this precipitation are not seen in the figure due to missing data in the next days.

RC2 Comment 16

Figure 7: If possible adjust the colour legend to rounded numbers rather than 3.16e+02, e.g. 0.5; 1; 5; . . .300. Also, if possible, add the numbers (15, 30, 45 %) of the source sensitivities onto the isolines of figure itself (this is quicker than having this written in the caption).

**Authors' response:** the numbers in the legend of the Figure 7 are now written out.

Please also note the supplement to this comment:
https://www.biogeosciences-discuss.net/bg-2018-426/bg-2018-426-RC2-supplement.pdf

**References**

Liu, X., Qi, Y., Dong, Y., Peng, Q., He, Y., Sun, L., Jia, J., and Cao, C. (2014). Response of soil N2O emissions to precipitation pulses under different nitrogen availabilities in a semiarid temperate steppe of Inner Mongolia, China. *Journal of Arid Land, 6*(4), 410-422. doi:10.1007/s40333-013-0211-x

Peng, Q., Qi, Y., Dong, Y., Xiao, S., and He, Y. (2011). Soil nitrous oxide emissions from a typical semiarid temperate steppe in inner Mongolia: effects of mineral nitrogen fertilizer levels and forms. *Plant and Soil, 342*(1), 345-357. doi:10.1007/s11104-010-0699-1

---

## Author Response (AR1)

**Peer-review of the manuscript 'Attribution of N2O sources in a grassland soil with laser spectroscopy based isotopocule analysis'**

**Table of contents:**

|    | a)   | Authors' response to Nathaniel Ostrom's review                                                                            |
|----|------|---------------------------------------------------------------------------------------------------------------------------|
| 5  | b)   | Authors' response to the review by the anonymous referee 2                                                                |
|    | c)   | Authors' response to the short comment by Klaus Schäfer1                                                                  |
|    | d)   | Revised manuscript with revision-traces                                                                                   |
|    | Not  | ice: For better readability, the authors' comments are highlighted in grey throughout the document. The revised manuscrip |
|    | wit  | h highlighted changes can be found in section d). There, the changes resulting from Nathaniel Ostrom's review are high    |
| 10 | ligh | ted with green font and the changes due to the anonymous referee's review are highlighted with red font. The reference    |
|    | add  | ed due to the short comment by Klaus Schäfer is highlighted with purple font. Referred page and line numbers were         |

**a) Manuscript review by Nathaniel Ostrom**

matched to the revised manuscript.

We would like to thank Nathaniel Ostrom for his efforts in reviewing our manuscript and for his comments and suggestions,
which have led to a valuable improvement of our manuscript. Below, the referee's comments (RC1) are displayed followed by the authors' response, which are highlighted in grey.

General comments:

The authors present an innovative and thorough assessment of the microbial origins of  $N_2O$  emissions from a grassland soil based on spectroscopic measurements. This data set is clearly an advance for the field as extensive time- series data sets for

20 the isotopic composition of  $N_2O$  are rare but very important to constrain the dynamic nature of  $N_2O$  production from soils. While I appreciate the thoroughness of the data set and interpretations, I have a few central concerns regarding calibration and with the assessment of microbial origins of  $N_2O$  described in the paper.

RC1 Comment 1:

Characterization of sample isotope values based on two isotopically characterized standards is certainly the minimum necessary. Critical is also that the range of isotope values of the samples be encompassed by the range in isotope values of the standards. The range in isotope values for  $\delta^{15}N^{\alpha}$  and  $\delta^{15}N^{\beta}$  is quite good (although I don't know what fraction of the samples lie outside these ranges) the range in  $\delta^{18}O$  values of the standards is small and does not well encompass the range in sample values. The issue with this is not a just with the precision (which can be determined) but with the accuracy; values outside the range of the standards cannot be considered accurate even if precise. My personal opinion is to not publish isotope values

30 outside the range of standards however, I appreciate that there may be a statistical approach for providing greater confidence in this situation.

Authors' response: one of the central concerns the referee points out is related to calibration of  $N_2O$  delta values, in particular the usage of a two-point calibration approach covering all measurement data. We acknowledge that this is of key importance to guarantee the accuracy of measurement data. In past years our laboratory has put significant efforts in the preparation of

35 standard gases covering the range of delta values from strongly 15N depleted to enriched and gases were anchored to the international isotope ratio scales in close collaboration with Sakae Toyoda & Naohiro Yoshida / TIT and Willi A. Brand / MPI-BGC. Based on these gases we demonstrated linearity of delta measurements by QCLAS versus calibration gases already in 2008 (Waechter et al., 2008) and to our knowledge are one of few laboratories, which implemented a two point calibration approach up to now, while most labs in the N2O isotope community report results using a one point (offset) correction. Our laboratory has organized an inter – laboratory comparison to improve the comparability of  $N_2O$  isotope results (Mohn et al., 2014) and participate in a similar very recent approach led by Nathaniel Ostrom (Ostrom et al., 2018). In both campaigns, the results reported by our laboratory were always in very good agreement with results reported by Tokyo Institute of Technology

5 as already stated in the manuscript. In addition, the N2O isotope community since recently has first reference materials, USGS 51 and 52 (Ostrom et al., 2018), but with limited coverage of  $\delta^{15}$ N and  $\delta^{18}$ O.

In summary, we appreciate the comments of the referee that a two-point calibration approach covering all measurement data should be implemented, but would like to state that this is currently confined by the non-availability of suitable reference materials and the adapted approach in this manuscript therefore presents current best practice. We added the following statement to page 5 Line 21 ff:

10

... [] measured to monitor the data quality (Table 2). While S1 and S2 cover the range of  $\delta^{15}N^{\alpha}$  and  $\delta^{15}N^{\beta}$  values of the sample gas, for  $\delta^{18}$ O this is currently confined by the non-availability of suitable standard gases. Nonetheless, the implemented calibration procedure presents current best practice in particular as the linearity of the delta scale for QCLAS measurements was demonstrated already in 2008 (Waechter et al., 2008).

15 RC1 Comment 2:

> While the authors present an extensive data set on the concentration and isotopic composition of  $N_2O$  in air much of the data set is derived from periods of low flux when the concentration of  $N_2O$  is only slightly above atmospheric. The calculation of soil-derived  $N_2O$  is obtained from a simple isotope mixing model (soil- derived mixing with atmospheric  $N_2O$ ). In this model, there is considerable error associated with the soil-derived isotope values at low N2O concentration (the error increases as the

20 concentration of N2O declines). Further, the error around each calculated soil-derived isotope value increases as the difference in the isotopic composition between soil and atmospheric N2O decreases. For this reason, in our laboratory, we generally don't publish data on samples in which the N2O concentration is less than 30% greater than atmospheric (although this is an arbitrary value). The Keeling-plot approach does provide greater accuracy and precision but it is critical that the authors (1) report the uncertainty obtained for all soil-derived isotope values, (2) provide a clear discussion of this issue, and (3) consider not report-25 ing isotope values for which the error is very large and a reasonable explanation for what the cut-off error should be.

Authors' response: the authors agree that the uncertainty of  $N_2O$  source signatures derived by the Keeling plot approach generally increases with decreasing share of source N2O and should be given to support data interpretation. This relationship was illustrated for a similar TREX-QCLAS setup by Wolf et al. (2015), indicating that with only 12 ppb increase in N2O an average standard error of 2.2 ‰, 1.4 ‰ and 1 ‰ for the SP,  $\delta^{15}N^{bulk}$  and  $\delta^{18}O$  isotopic source signatures can be achieved. The

- 30 main reasons for the robustness of the technique applied by Wolf et al. (2015) and in the presented manuscript are the high number of gas samples ( $10.9 \pm 0.9$  measurements within each of the Keeling plot analysis) and the high sensitivity and precision of TREX-QCLAS. Furthermore, the uncertainties of the Keeling plot results were retrieved with a Monte Carlo simulation using the error seen in the target gas measurements. Thus, uncertainties for all soil-derived isotope values are contained implicitly in the given values. Nonetheless, we agree, that the criterion used in the manuscript to differentiate between valid and
- invalid measurements, significant correlation (p-value < 0.05) between  $\delta^{15}N^{\alpha}$  and  $\delta^{15}N^{\beta}$ , might not be justified. Therefore, an 35 alternative criterion was implemented and the corresponding paragraph on page 5 Line 41 ff was adapted:

"This procedure yielded 30 Keeling plot derived source signatures. The uncertainty of the source signatures was assessed based on the measured isotope delta values and N2O concentrations using a Monte-Carlo model with 200 iterations. A benchmark value of 10 % for the SP standard deviation was chosen as a criterion to distinguish valid measurements, finally leading to 12

N2O accumulation events." 40

RC1 Comment 3:

15

30

There are two important assumptions in the graphical approach to tracing the microbial origins of N2O that is used by the authors: (1) that  $\delta^{18}$ O is a conservative tracer of origins and (2) that the kinetic isotope effects associated with production of N2O as well as reduction of N2O are constants. As I discuss below I question the validity of both of these assumptions. I will

5 acknowledge, however, that this is a common approach in the literature but ask that the authors, at least, discuss these issues and how variation in the kinetic isotope effects might impact their model outcomes.

Authors' response: the authors agree that  $\delta^{18}$ O-N2O is not only controlled by the origin of the oxygen atom in the N2O molecule (nitrate, nitrite, soil water or molecular O2), but mainly driven by oxygen exchange of reaction intermediates (nitrate, nitrite) with soil water, e.g. Lewicka-Szczebak et al. (2016, 2017). Thereby,  $\Delta\delta^{18}$ O (N2O/H2O) =  $\delta^{18}$ O-N2O -  $\delta^{18}$ O-H2O for

10 N2O from bacterial denitrification in incubated soils is considered as relatively stable, given high oxygen exchange rates (Lewicka-Szczebak et al., 2017). In this respect, we agree that the sentence on page 10 Line 24 - 25 "Unfortunately, the interpretation of  $\delta^{18}$ O-N2O is further complicated by oxygen exchange between NO3- and soil water (Well et al., 2008; Kool et al., 2011)." is misleading and was reformulated to:

" $\delta^{18}$ O-N2O of denitrification is affected by oxygen exchange between reaction intermediates (NO3-, NO2-) and soil water as a function of WFPS (Well et al., 2008; Kool et al., 2011)."

In addition, we will add the following sentence to page 12 Line 9 - 10 ff:

... using this approach reduces the uncertainty of the calculated relative contributions of the different processes as the boxes are used to span the mixing line.  $\Delta\delta^{18}O$  (N2O/H2O) for denitrification, is considered to be relatively stable (Lewicka-Szczebak et al., 2016), in particular under high WFPS associated with close to 100 % oxygen exchange between soil water and reaction intermediates (Kool et al., 2011).

20 intermediates (Kool et al., 2011).

In addition, we will complement the legend of Figure 8:

... (b) SP versus  $\Delta \delta^{18}$ O of soil-emitted N2O according to Lewicka-Szczebak et al. (2017), where  $\Delta \delta^{18}$ O = d18O-N2O - d18O-H2O, the isotope effect between soil water and formed N2O.  $\Delta \delta^{18}$ O for BD/ND is considered to be relatively stable, because of the high oxygen exchange between soil water and reaction intermediates at high WFPS.

25 We appreciate the comment by the referee that kinetic isotope effects (KIE) associated with N2O production / consumption are not constant, which is a common (and essential) simplification using the dual isotope mapping approach in a natural, mixed culture system. We would like to stress this fact and add the sentence on page 10 Line 27 ff:

... may be indicators for  $N_2O$  reduction (Koba et al., 2009). It has to be mentioned, however, that fractionation factors may deviate depending on environmental conditions (Koster et al., 2013) or even over the course of a single experiment due to multiple reaction steps involved (Haslun et al., 2018).

RC1 Comment 4, Page 5, Line 21: Was data corrected based on measured values for the Target gas?

Authors' response: no, target gas measurements were conducted using the same analytical routine and calibration procedures as applied for the sample gas measurements. Therefore, they were not used to correct data but to assess the overall TREX-QCLAS analytical performance, e.g. repeatability.

35 RC1 Comment 5 Page 5, Line 32: Was the concentration correction performed daily? Did the range in concentration standards encompass the range in sample concentrations observed?

Authors' response: the concentration correction was performed based on measurements of diluted S1, and was conducted once per 24 hours. The sample gas concentrations were mostly, i.e. 511 out of 612 samples, in the concentration range covered

by the concentration correction. The remaining 101 samples offered on average 5 % (maximum 14 %) higher concentrations, but were well within the linear range of the concentration correction. This linear dependency is typical for a delta correction at enhanced concentrations and a clear advantage of TREX-QCLAS versus QCLAS without preconcentration, which shows a very strong and non-linear concentration dependence.

5 RC1 Comment 6 Page 6, line 2: Did the Keeling plot approach include calculation of the uncertainty surrounding the isotope values for the soil derived  $N_2O$ ? Was this evaluated using the Monte Carlo model as described? At concentrations only slightly above ambient this error can be very large. We generally discard data with a  $N_2O$  concentration < 30% above atmospheric levels although the decision to discard data should probably be made on the basis of the magnitude of the uncertainty.

Authors' response: yes, a Monte Carlo model was applied to estimate the uncertainty of the N2O isotope source signatures.
In detail, the standard deviation of repeated target gas measurements, i.e. repeatability, was used as an estimate for the uncertainty of sample gas delta values and N2O concentrations. A detailed discussion on the selected filter criterion is given in the author's response on RC1 comment 2.

RC1 Comment 7 Page 6, Line 2: Why was a correlation between  $\delta^{15}N^{\alpha}$  and  $\delta^{15}N^{\beta}$  considered a criterion for a valid measurement? While unpublished, our data indicates that most of the variation in SP during N2O production from the cNOR enzyme

15 is driven by  $\delta^{15}N^{\beta}$ ; which suggests that a lack of correlation could be a normal feature of production from denitrification.

Authors' response: The authors agree that filtering of data based on a correlation between  $\delta^{15}N^{\alpha}$  and  $\delta^{15}N^{\beta}$  might not be justified for all source processes. Therefore, a different selection algorithm was implemented as detailed in the author's responses on RC1 comment 2.

RC1 Comment 8 Page 7, Line 11: It is not really necessary to specify the "hard disk" failure as the cause of missing data. Simply stating equipment failure is fine. This is also mentioned in the Figure 4 legend.

**Authors' response:** we would like to keep the statement in the text to highlight the fact that the data gap was not caused by a failure of the TREX-QCLAS, but we deleted the respective statement in the legend of Figure 4.

RC1 Comment 9 Page 7, line 32: Delete "during": word not needed.

Authors' response: we agree, done.

20

25 RC1 Comment 10 Page 9, Line 30: On what basis was an  $r^2$  of 0.2 similarly used to exclude data? This seems like a very poor degree of correlation.

Authors' response: we agree that an  $r^2$  of 0.2 is not sufficient to filter valid source signatures. Therefore a different selection algorithm was implemented as detailed in the author's responses on RC1 comment 2.

RC1 Comment 11 Page 9, Line 31: Perhaps "calculated" is a better word than "extracted".

30 Authors' response: we agree, done.

RC1 Comment 12 Page 9, line 32: What are "relatively large uncertainties"? It would be better to state the uncertainty in the text as well as in the figures. Perhaps it would be best to exclude data with large uncertainties; as long as some reasonable criteria can be established to exclude data beyond a certain uncertainty (perhaps based on the range in isotope values between the sources).

35 **Authors' response:** we agree, that a quantitative indication on the uncertainties in this period should be given in the text and have changed the respective sentence to:

"[..] i.e., the extracted isotopic composition of N2O emitted from soil ( $\delta^{15}$ Nbulk, SP,  $\delta^{18}$ O) derived from the Miller-Tans method showed relatively large uncertainties and amounted to 2.8 – 9.8 ‰, 2.3 – 10.6 ‰ and 4.6 – 12.9 ‰, respectively (shaded areas in Figure 5).

RC1 Comment 13 Page 9, line 36: Yes, the poorer data quality for  $\delta^{18}$ O is likely a consequence of the fact that the standards

- 5 do not encompass the data. Generally, standards must encompass the range in data to assure accuracy in any calibration relationship. The issue is that this is does not only result in poorer precision but, more importantly, results in poorer accuracy. The authors are presenting data for which the accuracy is questionable. If the  $\delta^{18}$ O values obtained are substantially outside the range of the standards the authors should give considerable thought as to whether they should be excluded.
- Authors' response: we agree to the referee comments, that a two-point calibration approach covering all measurement data should be implemented, but would like to state that this is currently confined by the non-availability of suitable reference materials. Therefore, the adapted approach presents current best practice. A detailed discussion of this aspect and text added to the manuscript is given in the author's responses on RC1 comment 1.

RC1 Comment 14 Page 10, line 32: I am not fond of the graphical approach used by Koba et al to apportion sources of  $N_2O$  largely because  $\delta^{18}O$  is a poor tracer of microbial origins. It is well known that oxygen exchanges with water during the

15 microbial production of N2O, which can alter  $\delta^{18}$ O values (e.g. Kool references). Further, this approach relies on limited data sets for  $\delta^{18}$ O generated by cultures of fungal denitrification, bacterial denitrification and nitrification. Given this, I suggest that the authors, at least, acknowledge the limitations of the graphical approach in assessment of the origins of N2O.

For  $\delta^{15}N$ , this approach also depends not only on the isotopic composition of the nutrient source but also the kinetic isotope effect (KIE) associated with N2O production. The KIE is more accurately described as a net isotope effect and can be highly

20 variable depending upon conditions in the soil. The authors need to consider and discuss the effect of variation in the KIE on their model outcomes.

Authors' response: the authors agree that the dual isotope mapping approach by Koba et al. (2009) and Lewicka-Szczebak et al. (2017) relies on a number of assumptions, which should be stated in the text. A detailed discussion of this aspect and text added to the manuscript is given in the author's responses on RC1 comment 3.

- 25 RC1 Comment 15 Page 11, Line 17: Here again a single value is chosen for the KIE associated with N2O reduction whereas there are many papers that show variation. In Jinuntuya-Nortman et al (2010) we demonstrate that the SP net isotope effect associated with N2O reduction can approach zero per mil and is inversely correlated with water filled pore space. The authors demonstrate that water filled pore space varies. What is the affect of a varying NIE on the model outcomes? Indeed their inference that low SP values occur at high WFPS is entirely consistent with the role of WFPS in masking expression of frac-
- 30 tionation during  $N_2O$  reduction and may not reflect variation in the proportion of  $N_2O$  derived from nitrification and denitrification. The estimates of the rates of  $N_2O$  reduction may not be meaningful if the net isotope effect for reduction varies (as it likely does).

Authors' response: we agree that a single value selected for the  $\epsilon$ (SP) is a simplification and  $\epsilon$ (SP) values might vary, e.g. depending on WFPS (Jinuntuya-Nortman et al, 2010) and N2O/(N2+N2O) product ratio (Lewicka-Szczebak et al., 2015). Therefore we added the following statement on page 11 Line 17 ff:

35

... in accordance with Ostrom et al. (2007). Using a single  $\epsilon$ (SP) value is a simplification, however, as fractionation factors might vary, e.g. depending on WFPS (Jinuntuya-Nortman et al, 2010) and N2O/(N2+N2O) product ratio (Lewicka-Szczebak et al., 2015).

RC1 Comment 16 Page 13, line 12: Given the issues raised above, I am not convinced by the author's statement that natural abundance isotope studies are "an effective way to disentangle  $N_2O$  production pathways". A frank assessment of the strengths and weaknesses of the approach is needed.

Authors' response: we agree and reformulated the mentioned statement to:

5 Our findings confirm that natural abundance isotope studies of  $N_2O$  provide a way to trace  $N_2O$  production / destruction pathways, in particular when combined with supportive parameters or isotope modelling approaches (Denk et al., 2017).

**References related to Nathaniel Ostrom's review**

35

- Haslun, J. A., Ostrom, N. E., Hegg, E. L., and Ostrom, P. H. (2018). Estimation of isotope variation of N2O during
   denitrification by Pseudomonas aureofaciens and Pseudomonas chlororaphis: implications for N2O source apportionment. *Biogeosciences*, 15(12), 3873-3882. doi:10.5194/bg-15-3873-2018
  - Kool, D. M., Wrage, N., Oenema, O., Van Kessel, C., and Van Groenigen, J. W. (2011). Oxygen exchange with water alters the oxygen isotopic signature of nitrate in soil ecosystems. *Soil Biology & Biochemistry*, 43(6), 1180-1185. doi:10.1016/j.soilbio.2011.02.006
- 15 Koster, J. R., Well, R., Tuzson, B., Bol, R., Dittert, K., Giesemann, A., Emmenegger, L., Manninen, A., Cardenas, L., and Mohn, J. (2013). Novel laser spectroscopic technique for continuous analysis of N2O isotopomers - application and intercomparison with isotope ratio mass spectrometry. *Rapid Communications in Mass Spectrometry*, 27(1), 216-222. doi:10.1002/rcm.6434
- Lewicka-Szczebak, D., Well, R., Bol, R., Gregory, A. S., Matthews, G. P., Misselbrook, T., Whalley, W. R., and Cardenas, L.
   M. (2015). Isotope fractionation factors controlling isotopocule signatures of soil-emitted N2O produced by denitrification processes of various rates. *Rapid Communications in Mass Spectrometry*, 29(3), 269-282. doi:doi:10.1002/rcm.7102
  - Lewicka-Szczebak, D., Dyckmans, J., Kaiser, J., Marca, A., Augustin, J., and Well, R. (2016). Oxygen isotope fractionation during N2O production by soil denitrification. *Biogeosciences*, *13*(4), 1129-1144. doi:10.5194/bg-13-1129-2016
- 25 Lewicka-Szczebak, D., Augustin, J., Giesemann, A., and Well, R. (2017). Quantifying N2O reduction to N-2 based on N2O isotopocules validation with independent methods (helium incubation and N-15 gas flux method). *Biogeosciences*, 14(3). doi:10.5194/bg-14-711-2017
- Mohn, J., Wolf, B., Toyoda, S., Lin, C. T., Liang, M. C., Bruggemann, N., Wissel, H., Steiker, A. E., Dyckmans, J., Szwec, L., Ostrom, N. E., Casciotti, K. L., Forbes, M., Giesemann, A., Well, R., Doucett, R. R., Yarnes, C. T., Ridley, A. R., Kaiser, J., and Yoshida, N. (2014). Interlaboratory assessment of nitrous oxide isotopomer analysis by isotope ratio mass spectrometry and laser spectroscopy: current status and perspectives. *Rapid Communications in Mass Spectrometry*, 28(18), 1995-2007. doi:10.1002/rcm.6982
  - Ostrom, N. E., Gandhi, H., Coplen, T. B., Toyoda, S., Böhlke, J. K., Brand, W. A., Casciotti, K. L., Dyckmans, J., Giesemann, A., Mohn, J., Well, R., Yu, L., and Yoshida, N. (2018). Preliminary assessment of stable nitrogen and oxygen isotopic composition of USGS51 and USGS52 nitrous oxide reference gases and perspectives on calibration needs. *Rapid Communications in Mass Spectrometry*, 32(15), 1207-1214. doi:doi:10.1002/rcm.8157
  - Waechter, H., Mohn, J., Tuzson, B., Emmenegger, L., and Sigrist, M. W. (2008). Determination of N2O isotopomers with quantum cascade laser based absorption spectroscopy. *Optics Express*, 16(12), 9239-9244. doi:Doi 10.1364/Oe.16.009239
- Well, R., Flessa, H., Xing, L., Ju, X. T., and Romheld, V. (2008). Isotopologue ratios of N2O emitted from microcosms with NH4+ fertilized arable soils under conditions favoring nitrification. *Soil Biology & Biochemistry*, 40(9), 2416-2426. doi:10.1016/j.soilbio.2008.06.003

**b) Manuscript review by anonymous referee 2**

We would like to thank the anonymous referee #2 for his detailed review of our manuscript and for his precious comments and suggestions, which have led to a clear improvement of our manuscript. Below, the referee's comments (RC2) are displayed followed by the authors' response, which are highlighted in grey.

**5 General observations and detailed comments:**

The paper by Ibraim and co-authors presents a novel technique to measure a suite of isotopic fingerprints of N2O using a fielddeployable device. Since N2O-emissions and their isotopic ratios vary spatially and temporally to a large degree, such an instrument is very useful to allow extent our knowledge on the processes driving these N2O emissions. The measurements presented in the paper demonstrate that the QCLAS instrumentation works well under field conditions and allows its imple-

- 10 mentation in further studies. This successful field deployment is certainly a central step after a yearlong construction and test phase in the laboratory and I congratulate the team for this effort. Likewise it is clear to the authors of this study that measuring the isotopic information of N2O is just one step, and to interpret and exploit the data requires extensive other knowledge, ranging from meteorological boundary conditions, soil analyses, to interpreting the application of manure on the sampling site. As further outlined below, some of these non-measurement aspects of the paper should be improved to gain more clarity for
- 15 the readers. Since the current measurement set up was not perfect, i.e. only nighttime measurements provide robust results, a lot can be learned from this measurement campaign for future field applications. At some places, the authors already suggested how these limitations could be overcome with a better design etc. I have the impression that these "lesson learned statements" could be extended to guide future measurement campaigns in this area. In general, the paper is clearly structured, the figures are mostly instructive and the text is written nicely. I therefore welcome this paper for final publication after my and the other
- 20 reviewers comments are included in the final version.

**RC2 Comment 1**

In section 4.4.6 (page 13) you discuss the influence of the manure application  $(12^{th} July)$  on the calculated source signatures of the emitted N2O. A causal link between excessive nitrogen addition on subsequent N2O emissions from the soil is to be expected and might be the case. However, looking at the flux time series in Fig. 3 it is equally clear that N2O fluxes rise after

- 25 intense rainfall events and this also fulfils expectation. The highest N2O emissions within the entire study follow the strong rainfall event end of July here without a manure application. However, the manure application on the 12th July is almost synchronous with the rainfall event; the farmer apparently waited for rain to apply the manure. My feeling is that the discussion in section 4.4.6 focusses too strongly on the manure application, while the more likely driver behind the rising N2O emissions (intense rainfall) is not really discussed equally. A second argument for the "heavy rain hypothesis" is also the wide footprint
- 30 of the isotope measurements. From Fig. 7 I get the impression that the largest fraction of the emissions stems from outside of the dashed rectangle where the site is located. Only 15 to 30% of the N2O emissions came from the local field where the manure was applied, while the largest fraction comes from an area of a few km distance. It might be that the other fields in the surroundings were also fertilized at the same time by the farmers and thus the De-Fen site is representative, but this information is missing. I might be wrong, but you might gain additional insight to view the N2O flux data and the isotopic signatures also
- 35 from this heavy rainfall point of view (likewise WFPS) and a wider, more realistic regional footprint. In this respect, you might also put less weight on the  $NH_4$  and  $NO_3$  soil extracted solute data because these data might be too local for the footprint of the measured  $N_2O$  isotope signatures.

Authors' response: The authors agree that N2O emissions and isotopic signatures are driven by both manure (substrate) addition and rainfalls (WFPS) and it might not be possible to disentangle both effects for the manure application on 12 July. The

40 effect of WFPS is explicitly discussed in the section 4.1 ("N2O fluxes and WFPS") as well as in the sections 4.4.3 and 4.4.5. Nevertheless, we acknowledge the reviewer's comment on the fact that the N2O emissions strongly raised after rainfall events since each of the three major rainfall events on 24 June, 12 July and 23 July was followed by a clear increase of the N2O emission rates, which is in agreement with literature (Peng et al., 2011; Liu et al., 2014). Unfortunately, for the two "rainfall - only" events no N2O source signatures could be retrieved: around 24th June the observed data did not allow for significant Keeling plot analysis, while on 23rd July the measurement campaign was terminated and the fractional measurement data did not allow calculation of N2O isotopic composition. We also agree that the footprint of the N2O isotope measurements goes beyond the fertilized plot. For this reason, we added a number of statements to section 4.4.6:

5

Page 13 line 23: After the manure application on 12 July and rainfall events in the days thereafter a strong ...

Page 13 Line 24: ... in the isotopic composition of the applied precursors, by an enhanced fractionation due to higher substrate availability or changes in process conditions (e.g. WFPS, see sections above).

Page 13 Line 31: ... during nitrification may have been used as substrate for denitrification, given the increase in WFPS due 10 to intensive rainfall events.

**RC2 Comment 2**

As suggested in your conclusions, it will be valuable in future measurement campaigns to also sample air from chambers that are more representative of the site. In other words, the precious isotope measurement time during the day or for meteorological situations that do not lead to sufficient N2O accumulation in the boundary layer could be better invested.

Authors' response: In our second field study, which took place between July and December 2017 in Central Switzerland, we 15 deployed automated flux-chambers at the day-time according to a specifically designed measurement schedule. This work is currently in preparation for publication.

**RC2 Comment 3**

20

Please mention the footprint shown in Fig. 7 earlier in the paper. It would be helpful for some readers (including me) to be aware that the actual footprint of the N2O isotope data is more extended than what is visualized in Fig. 1.

Authors' response: We agree and we add the following sentence to the end of the section 2.1.1, where the De-Fen site is characterized:

Page 3 line 33: "[...] respectively (Raiffeisen Laborservice, Ormont, Germany). The average footprint area for N2O flux and isotope measurements is given in Figure 7"

**25 RC2 Comment 4**

Page 1, Lines 33 and 37: (i) mentioning the sink term" while N2O reduction acted as a major sink" may not clear to all readers. Does this refer to a consumption of N2O produced in the soil itself or also for ambient atmospheric N2O, i.e. a net sink to the atmosphere? (ii) "N2O reduction to N2 largely dictated the isotopic composition of measured N2O." Does this statement refer to all measured isotope ratios; this statement seems very general.

**30 Authors' response:**

- (i) Page 1 line 37: In the given context our statement referred to soil produced  $N_2O$ . We cannot exclude reduction of ambient N2O but at least no net uptake of N2O was observed. To emphasize this fact we added the following statement: "[...] source for N2O, while N2O reduction acted as a major sink for soil produced N2O."
- (ii) The dual isotope mapping approach (Fig. 8) indicates that irrespective of the selected approach (SP versus  $\Delta \delta^{15}$ N or 35 SP versus  $\Delta\delta^{18}$ O) and scenario (1: reduction first, 2: mixing first) differences in SP to pure BD/ND were mostly controlled by N2O reduction. We therefore would like to keep this statement.

RC2 Comment 5

Page 3, Lines 32: stick to one name for the management "cutting" vs moving (caption Fig. 4)

**Authors' response:** Done. "Mowing" was changed to "cutting" at the following positions: page 6 line 40, page 7 line 6 and caption of Figure 4 at page 21 line 9.

**RC2 Comment 6**

5 Page 3, Line 30: Site name: Could you use just Fendt as the site name in your paper rather than the awkward De-Fen (I know the acronym De-Fen is the more official in terms of the European Flux Database cluster).

Authors' response: We would like to keep "De-Fen" for consistency with other work related to this research site.

**RC2 Comment 7**

Page 3, Lines 36: (i) I am not a specialist for agricultural manures, but my understanding is that manure usually refers to animal
feces (with the N mostly in form of urea) so I am confused by the ammonium N and referring to the Raiffeisen Laborservice;
Do you mean inorganic fertiliser e.g. pure Ammonium sulfate? In any case, please specify this. (ii) Further, I wonder if it
would have been worthwhile to obtain also the bulk N-isotopic composition of the two different kinds of fertilizers/manure.
You put a lot of effort into measuring the spatial and temporal distribution of the δ15N of soil-extracted nitrate while a value for the manure might be valuable as well for the input signature of δ15N of soil NH4

**15 Authors' response:**

- (i) The referee is correct, manure (and not ammonium sulphate) was applied. Raiffeisen Laborservice offers manure analysis (including analysis of N contents). Please see here: https://www.raiffeisen-laborservice.de/biogas/analysen-guelle.
- (ii) We agree that it would have been beneficial to get the manure analysed for the  $\delta^{15}N$  content, especially for the interpretation of the event around 12 July. Unfortunately, we did not sample the manure that was applied at De-Fen for subsequent analysis.

**RC2 Comment 8**

20

Page 4, Line 16: This measurement-specific information seems not necessary here ("While. . .) and could be deleted.

Authors' response: we agree. The sentence "While  $NH_4^+$  was converted to an indophenol complex,  $NO_3^-$  was reduced to nitrite to produce the diazo chromophore." was deleted.

**RC2 Comment 9**

Page 5, Line 6: sentence could be shortened: "Then the gas was dried using a Nafion dryer. . .(). . ..also, delete: overpressured (the 4.5 bar already indicate that)

Authors' response: The sentence was adapted as follows "Then the sample gas was dried using a nafion drier [...]". We would like to keep the term "sample gas" in order to emphasize the gas type. The term "overpressure", however, was deleted.

**RC2 Comment 10**

Page 6, Line 3: Given the complexity of the pathways, this correlation criterion is not a sound argument for a valid measurement as it discards 18 out of 30 values while accepting 12 only leading to a bias in the results.

Authors' response: The main reason for discarding many Keeling plot derived source signatures, e.g. in the period 10 June
to 21 June, was the low accumulation of N2O. However, since both reviewers commented on the selection criteria for valid source signatures, we adopted the criteria as stated in our comment 2 to reviewer 1.

RC2 Comment 11

Page 9, Line 1. You could end the sentence after the Toyoda citation and delete after ", but. . ." as this does not add much.

Authors' response: We would like to keep this part of the sentence to highlight differences between agricultural and suburban sites.

**RC2 Comment 12**

5

On page 9, line 4 you write: "At night, within a stable nocturnal boundary layer, vertical wind speeds and hence tracer transport are low, while lateral wind speeds can be high and constituents like  $N_2O$  can be transported over larger distances. As a result,  $N_2O$  emissions from other land uses or land cover may have contributed to the observed  $N_2O$  isotopic composition. To assess the possible influence of other land use / land cover. Please omit may and possible in these occasions where you actually know that more distant emission contribute.

Authors' response: We agree, done.

10 RC2 Comment 13

- (i) Table 1: unit for bulk density is not % but rather g/cm3; pH is dimensionless
- (ii) Table 2: to prevent confusion with the units, provide all values for the mole fraction in ppm, i.e. for T 0.329 ppm.
- (iii) Table 3: Event no (a.u)? did not get that for the three columns with SPKeeling and ïA, d'15N and ïA, d'18O: one digit seems enough for the +- values, i.e. 1.9 instead of 1.91 for SP.
- 15 (iv) Table 4: caption: better: Characterization of the lower and upper range for. . . first column header "Source signature" should read parameter or signature

**Authors' response:**

- (i) Done
- (ii) Done
- (iii) 'Event no. (a.u.)' was changed to 'Event number' and the number of digits was reduced to one as suggested
   (iv) We agree, done

**RC2 Comment 14**

- (i) Figure 3: Note, the x axis label for Fig. 3 and 4 and 5, 6 are all different (Date vs Datum in). Please select one for all, e.g. Date in 2016.
- 25 (ii) panel c with precipitation: 80 mm per hour seems a very high value, please check.
  - (iii) please rewrite sentence to omit, respectively: "Blue and red dashed lines refer to a cutting event and to a manure application, respectively." to: The blue dashed line indicates a cutting event and the red line manure application.
     (similar as you write in caption Fig. 4).

**Authors' response:**

- 30 (i) Done, 'Date in 2016 (dd.mm.)' was chosen for all
  - (ii) Done
  - (iii) Done

**RC2 Comment 15**

- (i) Figure 4: please explain the values given on the right side of the histograms
- 35 (ii) zoom panel a: at around 20.7. there is a weird magenta dot within the background values (black dot)
  - (iii) caption: please replace y-axis with axes (plural) general observation at Fig. 4: In Fig. 3 it becomes apparent that the heavy precipitation events (around 12.7. and 22.7.) that lead to a progressive reduction of the WFPS are strongly connected with two prominent N2O fluxes.

(iv) While the first heavy rain event (around 12.7.) is connected with the manure application, the second rainfall event happens without manure application. It would be worthwhile to add these heavy rainfall events also in Fig. 4 with lines or other markers.

**Authors' response: We agree and we will make the following changes:**

5 (i)

10

20

(ii) Done.

Done

- (iii) Done.
- (iv) The precipitation mentioned by the reviewer occurred on 23 July. As the scaling of Fig. 3 and 4 are different, the rainfall event occurs at the end of the N2O isotope measurements and did not show up in the results. We therefore prefer to not add a vertical line on the last day of this plot, because possible consequences of this precipitation are not seen in the figure due to missing data in the next days.
- RC2 Comment 16

Figure 7: If possible adjust the colour legend to rounded numbers rather than 3.16e+02, e.g. 0.5; 1; 5; ... 300. Also, if possible, add the numbers (15, 30, 45 %) of the source sensitivities onto the isolines of figure itself (this is quicker than having this

15 written in the caption).

Authors' response: the numbers in the legend of the Figure 7 are now written out.

**References related to anonymous referee's review**

Liu, X., Qi, Y., Dong, Y., Peng, Q., He, Y., Sun, L., Jia, J., and Cao, C. (2014). Response of soil N2O emissions to precipitation pulses under different nitrogen availabilities in a semiarid temperate steppe of Inner Mongolia, China. *Journal of Arid Land*, 6(4), 410-422. doi:10.1007/s40333-013-0211-x

Peng, Q., Qi, Y., Dong, Y., Xiao, S., and He, Y. (2011). Soil nitrous oxide emissions from a typical semiarid temperate steppe in inner Mongolia: effects of mineral nitrogen fertilizer levels and forms. *Plant and Soil*, 342(1), 345-357. doi:10.1007/s11104-010-0699-1

**c) Short comment by Klaus Schäfer**

- 25 A very different method to determine N2O emission fluxes was applied in Schäfer et al. (2012) to determine such fluxes from unfertilized grassland on the field scale. Fluxes in about the same amount, which are found and described here by Ibraim et al. in chapter 3.1 "N2O fluxes and soil parameters", were determined in our investigations by a tunnel, coupled to an open-path Fourier transform infrared spectrometer, which covered 500 m2, from 0 up to 14 µg N2O-N m-2 h-1. The FLEXPART-COSMO simulations had source sensitivity which originates from areas within approximately 300 m to 700 m distance to the
- 30 sample inlet representing an up-scaling similar to the tunnel method (100 m length). Peak emissions were detected in Schäfer et al. (2012) by concurrent chamber measurements after rainfall as described by Ibraim et al. also.

**Authors' response**

We agree that the two different approaches yielded similar results, which is very pleasant. Furthermore, while the FLEXPART-COSMO simulations were carried out to track the origin of the N2O measured with the TREX-QCLAS instrumentation, the

35 N2O fluxes described in chapter 3.1 were obtained with a measurement system consisting of coupled flux-chambers and GC-ECS. Since the agreement between the TREX-QCLAS method and the GC-ECD method was good (SI Figure 2), we may conclude that all approaches, i.e. including the tunnel method, agreed well. We appreciate the short comment by Klaus Schaefer and acknowledge that the study mentioned therein is closely related to our work and, thus, we will refer to this work in our reviewed manuscript.

**Attribution of N2O sources in a grassland soil with laser spectroscopy based isotopocule analysis**

Erkan Ibraim1,3, Benjamin Wolf2, Eliza Harris4, Rainer Gasche2, Jing Wei1, Longfei Yu1, Ralf Kiese2,
Sarah Eggleston1, Klaus Butterbach-Bahl2, Matthias Zeeman2, Béla Tuzson1, Lukas Emmenegger1, Johan Six3, Stephan Henne1, and Joachim Mohn1
1Empa, Swiss Federal Laboratories for Materials Science and Technology, Laboratory for Air Pollution & Environmental Technology, CH-8600 Dübendorf, Switzerland
2 Karlsruhe Institute of Technology, Institute of Meteorology and Climate Research (IMK-IFU), D-82467 Garmisch-Partenkirchen, Germany
 3ETH-Zürich, Swiss Federal Institute of Technology, Department of Environmental Systems Science, CH-8092 Zürich, Switzerland
 4University of Innsbruck, Institute of Ecology, Sternwartestrasse 15, A-6020 Innsbruck, Austria
 *Correspondence to*: E. Ibraim (erkan.ibraim@empa.ch)

**15 Abstract**

Nitrous oxide (N2O) is the primary atmospheric constituent involved in stratospheric ozone depletion and contributes strongly to changes in the climate system through a positive radiative forcing mechanism. The atmospheric abundance of N2O has increased from 270 ppb during the pre-industrial era to approx. 330 ppb in 2018. Even though it is well known that microbial processes in agricultural and natural soils are the major N2O source, the contribution of specific soil processes is still uncertain.

- 20 The relative abundance of N2O isotopocules (14N14N16N, 14N15N16O, 15N14N16O and 14N14N18O) carries process-specific information and thus can be used to trace production and consumption pathways. While isotope ratio mass spectroscopy (IRMS) was traditionally used for high-precision measurement of the isotopic composition of N2O, quantum cascade laser absorption spectroscopy (QCLAS) has been put forward as a complementary technique with the potential for on-site analysis. In recent years, preconcentration combined with QCLAS has been presented as a technique to resolve subtle changes in ambient N2O
- 25 isotopic composition.

From the end of May until the beginning of August 2016, we investigated N2O emissions from an intensively managed grassland at the study site Fendt in Southern Germany. In total, 612 measurements of ambient N2O were taken by combining preconcentration with QCLAS analyses, yielding  $\delta^{15}N^{\alpha}$ ,  $\delta^{15}N^{\beta}$ ,  $\delta^{18}O$  and N2O concentration with a temporal resolution of approximately one hour and precisions of 0.46 ‰, 0.36 ‰, 0.59‰ and 1.24 ppb, respectively. Soil  $\delta^{15}N$ -NO3- values and

- 30 concentrations of NO3- and NH4+ were measured to further constrain possible N2O-emitting source processes. Furthermore, the concentration footprint area of measured N2O was determined with a Lagrangian particle dispersion model (FLEXPART-COSMO) using local wind and turbulence observations. These simulations indicated that night-time concentration observations were largely sensitive to local fluxes. While bacterial denitrification and nitrifier denitrification were identified as the primary N2O-emitting processes, N2O reduction to N2 largely dictated the isotopic composition of measured N2O. Fungal
- 35 denitrification and nitrification-derived N2O accounted for 34 42 % of total N2O emissions and had a clear effect on the measured isotopic source signatures. This study presents the suitability of on-site N2O isotopocule analysis for disentangling source and sink processes in-situ and found that at the Fendt site bacterial denitrification/ nitrifier denitrification is the major source for N2O, while N2O reduction acted as a major sink for soil produced N2O.

**1. Introduction**

Nitrous oxide (N2O) is the third most important greenhouse gas (GHG), accounting for 6 % of the total anthropogenic radiative forcing (Myhre et al., 2013), and is thus far the dominant stratospheric ozone depleting substance emitted in the  $21^{st}$  century (Ravishankara et al., 2009). Its globally averaged atmospheric concentration has increased since the preindustrial era from

- 5 approximately 270 ppb (parts-per-billion,  $10^{-9}$  mole mole-1) at an average rate of 0.2 0.3% yr-1 and reached  $328.9 \pm 0.1$  ppb in 2016 (Prinn, 2016; WMO and GAW, 2016). While it is well known that natural and agricultural soils are the major N2O sources on a global scale, the relative contributions of individual microbial and abiotic N2O production and consumption pathways remain largely uncertain because different N2O-producing and -consuming processes are active simultaneously in a soil. Until now, there were no direct methods that allow for the attribution of an emitted amount of N2O to a given process in
- 10 the field (Solomon et al., 2007; Billings, 2008; Butterbach-Bahl et al., 2013). However, a detailed understanding of the temporal and spatial variations in N2O emissions and controlling processes is required to develop mitigation strategies and to better achieve emission reduction targets (Nishina et al., 2012; Cavigelli et al., 2012; Herrero et al., 2016; Decock et al., 2015). Atmospheric N2O isotopic composition provides important information about N2O production and consumption processes because distinct microbial and abiotic process pathways exhibit characteristic isotopic signatures (Toyoda et al., 2017; Decock
- 15 and Six, 2013b; Verhoeven et al., 2018; Denk et al., 2017). Apart from  ${}^{14}N^{14}N^{16}O$ , representing 99 % of total atmospheric N2O, the three most abundant N2O isotopocules are  ${}^{14}N^{15}N^{16}O$  ( ${}^{15}N$  at central  $\alpha$  position),  ${}^{15}N^{14}N^{16}O$  ( ${}^{15}N$  at terminal  $\beta$  position) and  ${}^{14}N^{14}N^{18}O$  (Toyoda and Yoshida, 1999; Kato et al., 1999). Abundances of isotopocules are usually reported in the  $\delta$ -notation in per mil (‰) as  $\delta^{15}N^{\alpha}$ ,  $\delta^{15}N^{\beta}$ ,  $\delta^{18}O$ , calculated according to the equation (1):

$$\delta X = (R_{\text{sample}} - R_{\text{standard}}) / R_{\text{standard}}$$
(1)

- where X denotes 15Nα, 15Nβ or18O and R refers to 14N15N16O / 14N14N16O, 15N14N16O / 14N14N16O or 14N14N18O / 14N14N16O, respectively, in a sample or standard (Toyoda and Yoshida, 1999; Brenninkmeijer and Röckmann, 1999; Werner and Brand, 2001). The international isotope reference scale for 15N / 14N is atmospheric N2 (AIR-N2) and for 18O / 16O Vienna Standard Mean Ocean Water (VSMOW). Thermal decomposition of isotopically characterized ammonium nitrate (NH4NO3) has been suggested as an approach to link the position-dependent nitrogen isotopic composition of N2O to AIR-N2 (Toyoda and Yoshida, 2001).
- 25 1999; Mohn et al., 2016). The total 15N content is usually reported as bulk 15N content ( $\delta^{15}N^{\text{bulk}}$ ) according to equation (2):

$$\delta^{15} \mathbf{N}^{\text{bulk}} = \left(\delta^{15} \mathbf{N}^{\alpha} + \delta^{15} \mathbf{N}^{\beta}\right) / 2 \tag{2}$$

while the site preference (SP) is used to denote the intramolecular 15N distribution according to the equation (3):

$$SP = \delta^{15} N^{\alpha} - \delta^{15} N^{\beta} \tag{3}$$

[revised manuscript text omitted]

**2.1.3 Concentrations of soil extracted NH4+and NO3- and $\delta^{15}$ N-NO3-**

Soil samples (approx. 150 g, 2-7 cm depth) were collected twice per week in a sampling grid (mesh size 70 m) spanning the whole measuring area at the De-Fen site (dashed square Fig. 1; Wolf et al., 2017), extracted with 1M potassium chloride (KCl, Merck KGaA, Darmstadt, Germany) and stored at -18 °C. After the manure application, sampling was increased to daily time

15 intervals (12 July 2016 – 15 July 2016), followed by further sampling on 19, 21 and 27 July 2016. The concentrations of  $NH_4^+$ and  $NO_3^-$  were determined colorimetrically using a spectrophotometer (AGROLAB Agrarzentrum GmbH, Germany). While  $NH_4^+$  was converted to an indophenol complex,  $NO_3^-$  was reduced to nitrite to produce the diazo chromophore.

[revised manuscript text omitted]
 ( $\mathbb{R}^2 > 0.2$  for  $\delta^{15} N^{\alpha}$  and  $\delta^{15} N^{\beta}$  versus
- 40 inverse concentration), and the source signatures, i.e., the extracted calculated isotopic composition of N2O emitted from soil

( $\delta^{15}N^{bulk}$ , SP,  $\delta^{18}O$ ) derived from the Miller-Tans method showed relatively large uncertainties and amounted to 2.8 – 9.8 ‰, 2.3 – 10.6 ‰ and 4.6 – 12.9 ‰, respectively (shaded areas in Figure 5) the extracted isotopic composition of N2O emitted from soil ( $\delta^{15}N^{bulk}$ , SP,  $\delta^{18}O$ ) derived from the Miller Tans method showed relatively large uncertainties (shaded area in Figure

 $\rightarrow$ . Thereafter, N2O source signatures as estimated with the Keeling plot and Miller–Tans approaches show a comparable trend

- 5 and mostly agree within the indicated uncertainties without systematic deviations. Overall, the agreement (R2 value) between the Miller–Tans and Keeling plot results is best for  $\delta^{15}N^{bulk}$  (0.84), intermediate for SP (0.57) and weakest for  $\delta^{18}O$  (0.39) (Figure 5). The weaker correlation for  $\delta^{18}O$ -N2O can be explained by a lower analytical data quality as compared to  $\delta^{15}N^{bulk}$ and SP, exemplified by a higher standard deviation for repeated measurements of the target gas (0.59 ‰ for  $\delta^{18}O$  and 0.41 ‰ for  $\delta^{15}N^{bulk}$  and SP). The reasoning behind this effect might be that the calibrated range of  $\delta^{18}O$  values (S1, S2) does not cover
- 10 the isotopic composition of the target and sample gases, because no suitable calibration gas was available. A difference of 7 % in  $\delta^{18}$ O between the two calibration gases is rather small, leading to a relatively high uncertainty in the respective calibration factors.

The base calculation for both the Keeling plot and Miller-Tans is identical and the two methods would yield identical results if every term was known perfectly. However, the uncertainty term is treated differently in the two approaches. The Miller-

- 15 Tans approach calculates source signatures for individual sample gas measurements (SI Figure 3) and, thus, may be the better choice when the source process or the background N2O isotopic composition changes rapidly, i.e. during a 24 hour period. However, the large fluctuations of the source signatures (up to 100 ‰, Figure 5) extracted by the Miller-Tans approach prior to 22 June indicate that the uncertainty estimated for the Miller-Tans approach is too optimistic and needs to be reassessed. In addition, it is noteworthy that the Keeling plot approach as presented here, implicitly considers changes in background N2O
- 20 concentration from day to day, since one Keeling plot (comprising both N2O background and N2O variations) was carried out per day. Therefore, we conclude that the Keeling plot method remains a robust way of estimating source signatures of N2O emitted from a predominantly agricultural landscape as the one presented here, where variations in background N2O compared to source contributions can be neglected and changes in source processes generally occur only on long timescales as a response to changes in environmental conditions (e.g. WFPS).

**25 4.4.2 Range of N2O source signatures**

Typical source signatures of biologically produced N2O are approx. -40 - 0 ‰ and 0 - 40 ‰ for  $\delta^{15}$ Nbulk and SP, respectively, while  $\delta^{18}$ O-N2O are around 40 ‰ and 70 ‰ for N2O emitted through grasslands or wetlands, respectively (Toyoda et al., 2017). Accordingly, the  $\delta^{15}$ Nbulk values found in our study are well within literature values of grassland emitted N2O, while the SP values are rather high. Interestingly, the obtained  $\delta^{18}$ O values were strongly elevated on some occasions and close to

- 30 those found by Ostrom et al. (2007) in a pure culture experiment in which approx. 80 % of produced N2O was reduced to N2. A correlated increase of the SP,  $\delta^{15}N^{bulk}$  and  $\delta^{18}O$  values, with SP values potentially larger than the endmember value of 32.8  $\pm$  4‰, can be explained by N2O reduction to N2, which is particularly active under wet and anaerobic soil conditions (Wrage et al., 2004; Lewicka-Szczebak et al., 2017). Thus, isotopic fractionation during partial N2O reduction must be taken into account in order to apportion isotopic source signatures of soil-emitted N2O (Lewicka-Szczebak et al., 2017; Verhoeven et al.,
- 35 2018). The fractionation factors  $\varepsilon^{18}O/\varepsilon^{15}N^{bulk}$ ,  $\varepsilon^{18}O/\varepsilon SP$  and  $\varepsilon^{15}N^{bulk}/\varepsilon SP$  have been determined in a number of incubation experiments and it has been suggested that their ratios (2.4, 2.8 and 1.2, respectively) may be indicators for N2O reduction (Koba et al., 2009). It has to be mentioned, however, that fractionation factors may deviate depending on environmental conditions (Koster et al., 2013) or even over the course of a single experiment due to multiple reaction steps involved (Haslun et al., 2018). Furthermore,  $\delta^{18}O-N_2O$  of denitrification is affected by oxygen exchange between reaction intermediates (NO3-,
- 40 NO2) and soil water as a function of WFPS Unfortunately, the interpretation of  $\delta^{48}$ O N2O is further complicated by oxygen exchange between NO3 and soil water (Well et al., 2008; Kool et al., 2011).

**4.4.3 N2O source partitioning using SP and $\Delta \delta^{15} N^{bulk}$**

An SP-versus- $\Delta \delta^{15}$ Nbulk (Figure 8(a)) mapping approach as originally presented by Koba et al. (2009) was used to interpret the Keeling plot-derived source signatures with respect to the possible underlying N2O producing and consuming processes. Here,  $\Delta \delta^{15}$ Nbulk denotes the  $\delta^{15}$ N difference between the product N2O and its substrate (NO3-). While Koba et al. (2009) applied this

- 5 approach in the framework of a groundwater study where  $NO_3^-$  was the only available N2O substrate, the grassland research site De-Fen showed rather high NH4+ concentrations (Figure 3(b)). Therefore, the N2O substrate at De-Fen might be either NH4+ for N2O emitted by nitrification (N) and nitrifier-denitrification (ND) or NO3- from fungal denitrification (FD) and bacterial denitrification (BD). Within the framework of this study, it was assumed that  $\delta^{15}N-NH_4^+$  and  $\delta^{15}N-NO_3^-$  values were in a similar range, i.e. approx. 0 – 15 ‰, in agreement with the literature (Mook, 2002; Holland, 2011). We thus used only the
- 10  $\delta^{15}N-NO_3^-$  values for the substrate isotopic composition. For periods where N2O emissions were present but no  $\delta^{15}N-NO_3^-$  values were obtained, the  $\delta^{15}N-NO_3^-$  values were approximated by linear interpolation. In addition, the concept of Koba et al. (2009) was modified for the two N2O-emitting domains FD/N and BD/ND using literature values as provided in Table 4. For simplicity, in the remaining part of this section the flux-weighted average values of SP and  $\Delta\delta^{15}N^{bulk}$  are discussed, while values of individual events can be found in Table 4. Plotting SP vs.  $\Delta\delta^{15}N^{bulk}$  revealed that there was a trend of increasing SP
- 15 with decreasing  $\Delta \delta^{15}$ Nbulk values. As indicated in Figure 8 with orange crosses, the flux-averaged SP,  $\Delta \delta^{15}$ Nbulk and  $\delta^{18}$ O values were 23.4, 19.0 and 62.3 ‰, respectively. The slope of the SP-versus- $\Delta \delta^{15}$ Nbulk linear regression line of -0.85 (solid red arrow in Figure 8(a)) is in agreement with literature values (-0.83 and -1.1) given by Koba et al. (2009) and Toyoda et al. (2017) for partial N2O-to-N2 reduction. This observed negative slope, which is in contrast to the grey shaded area anticipated for mixing of N2O produced by BD/ND and FD/N indicates a major contribution of BD/ND and N2O reduction to N2, the final reaction
- 20 step in the anoxic reduction of  $NO_3^-$  to  $N_2$ . The suspected predominance of denitrification agrees with previous field studies presented by Opdyke et al. (2009), Wolf et al. (2015) and Mohn et al. (2012). SI Figure 9 illustrates contributions of FD/N on the total N2O emissions for individual accumulation events.

A semi-quantitative source partitioning can be calculated assuming average SP (-0.9 ‰) and  $\Delta \delta^{15}N^{bulk}$  (18.5 ‰) values for N2O production by BD/ND and a fixed SP/  $\Delta \delta^{15}N^{bulk}$  ratio of -0.83 for N2O reduction to N2 (Figure 8(a)). Correspondingly,

- 25 the simultaneous SP increase and  $\Delta \delta^{15}N^{bulk}$  decrease during N2O reduction to N2 can be interpreted in terms of the N2O/ (N2O+N2) product ratio using the Rayleigh fractionation approach of Mariotti et al. (1981). Accordingly, a 90 % reduction of N2O translates into an increase in SP by 13.6 ‰ assuming an SP fractionation factor ( $\epsilon$ SP) of -5.9 ‰ in accordance with Ostrom et al. (2007). Using a single  $\epsilon$ (SP) value is a simplification, however, as fractionation factors might vary e.g. depending on WFPS (Jinuntuya-Nortman et al, 2010) and N2O/(N2+N2O) product ratio (Lewicka-Szczebak et al., 2015). A deviation of
- source signatures from the SP/ Δδ15Nbulk line can then be interpreted in terms of addition of N2O produced by additional processes, e.g. FD/N. This interpretation is supported by the relationship between SP and WFPS (Figure 6). Accordingly, the lowest SP values were found at intermediate to high soil water contents (80 90 % WFPS) along with maximum N2O fluxes, while SP values increased towards lower WFPS values, due to the increasing contribution of nitrification, and towards higher WFPS values, due to increasing N2O reduction to N2. Furthermore, the fraction of FD/N-derived N2O increased with NH4+
  fertilization, also in agreement with the literature (Tovoda et al., 2011; Koster et al., 2011).
- 35 fertilization, also in agreement with the literature (Toyoda et al., 2011; Koster et al., 2011). A semi-quantitative interpretation of isotope signatures can be done assuming average source signature values (SP and Δδ15Nbulk) and considering two scenarios (see also SI Figure 4): in scenario 1, BD/ND-produced N2O is partially reduced to N2 and the residual N2O (rN2O; remaining N2O after N2O reduction to N2) is then mixed with N2O 
[revised manuscript text omitted]
^{15} \mathrm{N}^{lpha}$ (‰) | $\delta^{15} \mathrm{N}^{eta}$ (‰) | $\delta^{18} { m O} \ (\%)$ | N 2 O mole fraction (*ppm) |  |  |
|------------|-------------------------------------|------------------------------------|-----------------------------|---------------------------------------|--|--|
| S 1 | $15.51 \pm 0.30$                    | $-3.25 \pm 0.20$                   | 34.97 ± 0.16                | $90.15 \pm 0.005$                     |  |  |
| S2         | $\textbf{-63.08} \pm 0.78$          | $\textbf{-59.81} \pm 0.48$         | $27.99\pm0.28$              | $90.84\pm0.024$                       |  |  |
| Т          | $15.25\pm0.09$                      | $-3.37\pm0.13$                     | $43.80\pm0.17$              | 329.250.329 ± 1.010.001 |  |  |

\*Values of S1 and S2 given in ppm, value of T given in ppb

Table 3 Characterization of the accumulation events. Columns refer to date, water filled pore space (WFPS), observed N2O fluxes ( $f(N_2O)_{GC-ECD}$ ), Keeling plot-derived SP values, obtained net isotope effect for  $\Delta\delta^{15}N(NO_3-N_2O)$ , obtained net isotope effect for  $\Delta\delta^{18}O(N_2O/H_2O)$ , fraction of remaining N2O after N2O reduction (rN2O. sc11 = SP vs.  $\Delta\delta^{15}N(NO_3-N_2O)$ ) approach scenario 1, sc12 = SP vs.  $\Delta\delta^{15}N(NO_3-N_2O)$  approach scenario 2, sc21 = SP vs.  $\Delta\delta^{18}O(N_2O/H_2O)$  approach scenario 1, and sc22 = SP vs.  $\Delta\delta^{18}O(N_2O/H_2O)$  approach scenario 2). Results are sorted by descending WFPS.

| 5
Event <del>no.</del>
(a.u.) num-
ber | WFPS
(%)    | Date (2016)
(dd.mmm.) | f(N 2 O) GC-ECD
(μg N m -2 h -1 ) | SP Keeling
(‰)            | $\Delta \delta^{15} N(NO_3^ N_2O)$ (‰) | $\Delta \delta^{18} O(N_2 O/H_2 O)$ (‰) | rNit1
(a.u.) | rNit2
(a.u.) | rN2O_sc11
(a.u.) | rN2O_sc12
(a.u.) | rN2O_sc21
(a.u.) | rN2O_sc22
(a.u.) |
|---------------------------------------------------------------|----------------|--------------------------|----------------------------------------------------------------------------------|-----------------------------------------|----------------------------------------|-----------------------------------------|-----------------|-----------------|---------------------|---------------------|---------------------|---------------------|
| 1                                                             | $98.2\pm0.2$   | 22.Jun.                  | 50                                                                               | 31.9 ± 1.9 <mark>4</mark>               | $5.3\pm6.8\textcolor{white}{1}$        | $74.2 \pm 2.61$                         | 0.38            | 0.43            | < 0.01              | 0.03                | < 0.01              | 0.13                |
| 2                                                             | $97.2\pm0.6$   | 23. Jun.                 | 147                                                                              | $36.7 \pm 2.0$ 2                        | 14.4 ± 4.6 <mark>0</mark>              | $73.0 \pm 2.3$                          | 0.50            | 0.58            | < 0.01              | 0.04                | < 0.01              | 0.18                |
| 3                                                             | $95.3\pm0.9$   | 24. Jun.                 | 318                                                                              | $22.4 \pm 4.20$                         | $11.3 \pm 5.34$                        | $51.3 \pm 7.0$                          | 0.36            | 0.36            | 0.04                | 0.10                | 0.14                | 0.41                |
| 4                                                             | $93.2\pm0.8$   | 17. Jul.                 | 259                                                                              | 8.4 ± 3.7 <del>2</del>                  | 24.6 ± 6. <del>29-</del> 3             | 65.6 ± 3. <del>36-4</del>               | 0.35            | 0.25            | 0.76                | 0.81                | 0.11                | 0.21                |
| 5                                                             | $90.2 \pm 1.2$ | 18. Jul.                 | 260                                                                              | 34.3 ± 5.2 <del>0</del>                 | $18.7\pm6.34$                          | 70.7 ± 2. <del>09-1</del>        | 0.51            | 0.53            | < 0.01              | 0.07                | < 0.01              | 0.19                |
| 6                                                             | 87.7 ± 12.6    | 12. Jul.                 | 236                                                                              | $13.0 \pm 4.46-5$                       | 25.9 ± 4. <del>06-1</del>       | 46.6 ± 3. <del>37-4</del>               | 0.37            | 0.14            | 0.51                | 0.64                | 0.26                | 0.36                |
| 7                                                             | $85.5\pm2.7$   | 19. Jul.                 | 560                                                                              | $25.7\pm0.\textcolor{red}{\textbf{7}8}$ | $16.9 \pm 4.62$                        | $74.5\pm2.7\textcolor{white}{1}$        | 0.41            | 0.24            | 0.02                | 0.12                | 0.02                | 0.01                |
| 8                                                             | $73.8\pm9.3$   | 20. Jul.                 | 411                                                                              | 16.7 ±
<del>1.992.0</del>     | 18.7 ± 5. 6 59                  | 56.7 ± 2.48- 5                   | 0.36            | 0.15            | 0.16                | 0.28                | 0.12                | 0.21                |
| 9                                                             | $70.3\pm1.6$   | 09. Jul.                 | 596                                                                              | $29.6 \pm 1.0 \textcolor{white}{4}$     | 11.5 ± 5.0 4.95                 | $70.7 \pm 3.56$                         | 0.40            | 0.39            | 0.01                | 0.06                | 0.01                | 0.15                |
| 10                                                            | $66.0\pm1.9$   | 21. Jul.                 | 475                                                                              | 33.2 ± 7. <del>86-9</del>        | 21.1 ± 3. 9 85                  | $62.3 \pm 6.61$                         | 0.53            | 0.58            | < 0.01              | 0.10                | < 0.01              | 0.32                |
| 11                                                            | $64.1 \pm 2.1$ | 10. Jul.                 | 340                                                                              | 33.0 ± 3.9 <del>0</del>                 | 10.1 ± 5.0 4.97                 | $60.0\pm2.04$                           | 0.42            | 0.60            | < 0.01              | 0.04                | < 0.01              | 0.38                |
| 12                                                            | $58.8 \pm 1.3$ | 11. Jul.                 | 251                                                                              | $15.7\pm0.74$                           | $29.5\pm4.8\textbf{-}$                 | 55.8 ± 1. <del>06-1</del>        | 0.41            | 0.13            | 0.53                | 0.70                | 0.14                | 0.21                |

Table 4 Characterization of the lower and upper range for Characterization of lower and upper SP,  $\Delta \delta$ 15N and  $\Delta \delta$ 18O <del>boundaries</del> for the two N2O-emitting domains fungal denitrification/ nitrification (FD/N) and bacterial denitrification (nitrifier denitrification (BD/ND) according to literature. All values are given in per mil (‰).

[revised manuscript text omitted]

---

## Referee Report (RR1)

Review of Ibraim et al: "Attribution of N2O sources in a grassland soil…"
       By Nathaniel E. Ostrom, Michigan State University

The authors have carefully addressed the issues raised in review and overall this is an excellent manuscript with a compelling data set. I remain concerned with three points that were raised in my initial review.

1. Reporting of isotope standards outside the range of the standards. There is no doubt that the authors have an extensive history in the development of N2O standards and international collaboration to attain that goal. And I agree that the authors are conducting the "current best practice(s)" in the field. But this does not avoid the issue that, particularly with regard to d18O, the sample values are well outside the range of the standards and, further, the range in d18O values for standards is quite small.  The statement that "… the linearity of the delta scale for QCLAS measurements was demonstrated already in 2008 (Waechter et al., 2008)" is not very satisfactory as spectroscopic instruments are very susceptible to drift and require "frequent calibration" (Waechter et al., 2008). Further, linearity does not equate to accuracy. In that study, the d18O values for the standards ranged about 100 per mil vs only a few per mil in this study. As in archery, accuracy becomes much more difficult the further from the target you are. I agree that international standards are not available in this range, but Waechter et al. clearly demonstrated that working standards can be generated (although I expect that there is some uncertainty in the accuracy of those standard values). Thus, the authors are following best practices, but the standard should not be best practices, but do we have sufficient confidence in the accuracy of the sample values to publish? Calibration is far behind the applications and this is largely because of the difficulty in developing standards, lack of support from funding agencies and insufficient attention by the scientific community. I, for one, have chosen not to publish d18O values generated spectroscopically because of the limited range of values in our standards. I believe the isotope values I obtain are accurate, but the problem is that I don't know for certain. Further, the casual reader is very unlikely to be able to make this distinction. I honestly don't know the solution for this problem and have no desire to hold up publication. My leaning, as indicated, is not to publish the d18O values but, at the least, the authors should indicate that many isotope values are outside the range of the standards which can be problematic to precise calibration.

2. The "benchmark value of 10 per mil for the SP standard deviation" seems rather large given that the difference between microbial sources of N2O is about 30-40 per mil. As long as the authors are clear about this and, perhaps, report the standard deviations the reader will have the ability to evaluate accuracy.

3. I agree that given full exchange between N2O and water during bacterial denitrification the d18O values for N2O should be constant. I find the word "stable" is this context to be less accurate than "constant". But, what can we expect the d18O values to be when exchange is not 100% and how often does this happen?

---

## Author Response (AR2)

**Reply letter to Nathaniel Ostrom's second review of our manuscript bg-2018-426**

We would like to thank Nathaniel Ostrom for re-reviewing our manuscript after our manuscript revisions according to his previous suggestions and according to the anonymous referee's suggestions. Below, the referee's comments (RC1) are displayed followed by the authors' response, which are highlighted in grey.

General comments:

The authors have carefully addressed the issues raised in review and overall this is an excellent manuscript with a compelling data set. I remain concerned with three points that were raised in my initial review.

RC1 Comment 1:

Reporting of isotope standards outside the range of the standards. There is no doubt that the authors have an extensive history in the development of $N_2O$ standards and international collaboration to attain that goal. And I agree that the authors are conducting the "current best practice(s)" in the field. But this does not avoid the issue that, particularly with regard to $\delta^{18}O$, the sample values are well outside the range of the standards and, further, the range in $\delta^{18}O$ values for standards is quite small. The statement that "… the linearity of the delta scale for QCLAS measurements was demonstrated already in 2008 (Waechter et al., 2008)" is not very satisfactory as spectroscopic instruments are very susceptible to drift and require "frequent calibration" (Waechter et al., 2008). Further, linearity does not equate to accuracy. In that study, the $\delta^{18}O$ values for the standards ranged about 100 per mil vs only a few per mil in this study. As in archery, accuracy becomes much more difficult the further from the target you are. I agree that international standards are not available in this range, but Waechter et al. clearly demonstrated that working standards can be generated (although I expect that there is some uncertainty in the accuracy of those standard values). Thus, the authors are following best practices, but the standard should not be best practices, but do we have sufficient confidence in the accuracy of the sample values to publish? Calibration is far behind the applications and this is largely because of the difficulty in developing standards, lack of support from funding agencies and insufficient attention by the scientific community. I, for one, have chosen not to publish $\delta^{18}O$ values generated spectroscopically because of the limited range of values in our standards. I believe the isotope values I obtain are accurate, but the problem is that I don't know for certain. Further, the casual reader is very unlikely to be able to make this distinction. I honestly don't know the solution for this problem and have no desire to hold up publication. My leaning, as indicated, is not to publish the $\delta^{18}O$ values but, at the least, the authors should indicate that many isotope values are outside the range of the standards which can be problematic to precise calibration.

**Authors' response:** As also mentioned in our previous response, we do agree that calibration of $\delta^{18}O$-$N_2O$ data should be improved in future studies, given the availability of appropriate standard gases. As acknowledged by Nathaniel Ostrom, the two point calibration approach presented here corresponds to current best practice, while most other laboratories publish $\delta^{18}O$, but also $\delta^{15}N^\alpha$ and $\delta^{15}N^\beta$ results using a one point calibration approach based on a singular working standard. While recently available $N_2O$ calibration standards bracket a range of approx. 0.6 ‰ (Ostrom et al., 2018) and therefore do not enable two-point calibration, the standards S1 and S2 used in this study span approx. 7 ‰. The accuracy of our $\delta^{18}O$-$N_2O$ measurement data for samples outside the calibrated range was demonstrated in Mohn et al. (2014) and results of our laser spectroscopic technique were comparable to IRMS performance. We therefore would expect all users of laser spectroscopic as well as mass spectrometric techniques to implement a two point calibration approach, covering all measurement data. Nonetheless we agree to add a further statement to the conclusion section.

Page 13 Line 40: "Replicate in-situ measurements of a pressurized air tank demonstrated a short term repeatability of the TREX-QCLAS system of 0.25 ‰, 0.31 ‰, 0.30 ‰ and 0.25 ppb for $\delta^{15}N^{\alpha}$, $\delta^{15}N^{\beta}$, $\delta^{18}O$ and $N_2O$ concentration, respectively. Accuracy of results was assured using a two point calibration that entirely spanned the range of obtained $\delta^{15}N^{\alpha}$ and $\delta^{15}N^{\beta}$ values, however, did not fully cover the range of obtained $\delta^{18}O$ values. This is current best practice, as no suitable reference gases are available, but might lead to a somewhat larger uncertainty for $\delta^{18}O$-$N_2O$."

RC1 Comment 2:

The "benchmark value of 10 per mil for the SP standard deviation" seems rather large given that the difference between microbial sources of $N_2O$ is about 30-40 per mil. As long as the authors are clear about this and, perhaps, report the standard deviations the reader will have the ability to evaluate accuracy.

**Authors' response:** Actual standard deviations for reported SP values are given in Table 3 and range between 0.7 and 7.9 ‰, with $3.2 \pm 2.2$ ‰ (mean $\pm$ 1SD). Accordingly, the highest standard deviation actually presented in our manuscript corresponds to 7.9 ‰, while most of the them are much lower. Therefore, the reader has the ability to evaluate the uncertainty of the results and no additional measures are taken.

RC1 Comment 3:

I agree that given full exchange between $N_2O$ and water during bacterial denitrification the $\delta^{18}O$ values for $N_2O$ should be constant. I find the word "stable" is this context to be less accurate than "constant". But, what can we expect the $\delta^{18}O$ values to be when exchange is not 100% and how often does this happen?

**Authors' response:** We agree that "constant" is a better expression than "stable" in this context and replace "stable" by "constant" on page 12 line 9 and page 20 line 18.

The authors agree that the effect of oxygen exchange on the $\delta^{18}O$-$N_2O$ values is complex and depends on several parameters ($\delta^{18}O$ values of precipitation, $\delta^{18}O$ values of soil water, degree of oxygen exchange), which makes a quantification challenging.

**References**

[revised manuscript text omitted]